# Targeted imaging of specialized plant cell walls by improved cryo-CLEM and cryo-electron tomography

J. Daraspe[1,4], E. Bellani [2,4], D. De Bellis [1,2], C. Genoud [1,3,5] ✉ & N. Geldner [2,5] ✉

Cryo-focused ion beam scanning electron microscopy (Cryo-FIBSEM) has become essential for preparing cryo-lamellae. Here, we present a series of improvements that speed up and enhance the efficiency of the Serial Lift-Out and SOLIST (Serialized On-grid Lift-In Sectioning for Tomography) procedures. We extend the cryo-FIBSEM session to 5 days and eliminate the need of copper or gold block intermediates and reduce curtaining effects. Finally, we report a routine to target lamellae with a precision of approximately 1 μm in X/Y/Z. We demonstrate the power of our improvements by targeting Casparian strips, suberin lamellae, as well as xylem vessels of *Arabidopsis thaliana* roots. This requires reaching a target of 5 micrometers in a 3 mm long and 80-120 μm thick root section. Despite ice formation in vacuoles and to some degree in the cytosol, plasma membranes and cell walls are remarkably well preserved, providing stunning insights into the native, hydrated nano-structure of plant cell walls, previously only observable with contrasting agents in a dehydrated state.

Cryo-electron microscopy (cryo-EM) has revolutionized structural biology, allowing to study molecular complexes at near-atomic resolution[1]. There is now a shift in focus toward cryo-electron tomography (cryo-ET)[2], as it allows to image cellular ultrastructure in its most native state. At the same time, it can enable imaging of macromolecules within intact cells or tissues by capturing a series of tilted projections and reconstructing three-dimensional volumes[2,3].

Molecules can be directly imaged on grids in the cryo-transmission electron microscope. Larger samples, by contrast, stop electrons and cannot be imaged without being thinned[3]. Preparing thin, electron-transparent sections from complex samples remains challenging. Cryo-ultramicrotomy[4], as developed by Dubochet et al., has existed for 40 years, but is difficult to apply to most tissues[5]. Furthermore, targeting small and cell-type specific regions, such as Casparian strips in plant roots, is so difficult and prone to chance that it has never been done. Cryo-FIBSEM has been developed to cut and thin

cryo-lamellae on the grid on which the sample is frozen. This first approach has allowed for to directly thin out of adherent cells, bacteria, and yeasts successfully[6]. Yet, plunge-freezing is limited to thin samples below a couple of micrometers. Any more complex tissues, organoids, or large cells cannot be frozen and thinned out directly on the grid. The only method available to vitrify larger samples is high-pressure freezing (HPF). It requires placing a small piece of tissue (less than 200 μm thick) inside a carrier and insert it into a vitrifying instrument. In such cases, the final product is a piece of vitrified tissue inside a carrier, meaning it is not suitable to be used directly in cryo-TEM. The waffle method[7] consists of placing the sample on a grid inside the carrier before freezing. The grid is then collected and placed into a cryo-FIBSEM for thinning to obtain cryo-lamellae. While this approach is effective for specimens smaller than the grid bar thickness (typically 20–25 μm) and can accommodate slightly larger samples with the addition of a spacer, it presents significant limitations for

[1]Electron Microscopy Facility, University of Lausanne, Lausanne, Switzerland. [2]Department of Plant Molecular Biology, University of Lausanne, Lausanne, Switzerland. [3]Institute of Bioengineering, Life Science, EPFL, Lausanne, Switzerland. [4]These authors contributed equally: J. Daraspe, E. Bellani. [5]These authors jointly supervised this work: C. Genoud, N. Geldner. ✉e-mail: christel.genoud@unil.ch; niko.geldner@unil.ch

larger organisms or tissue samples. It is not suitable for our work with extensive root systems, as the dimensional constraints of the grid would introduce sample damage. Recent advances, such as Serial Lift-Out[8] and SOLIST[9] have improved throughput through innovative milling and lamella attachment strategies[10]. This workflow allows high-pressure freezing of pieces of tissue in a carrier and then transfer a region of interest on a grid to carve cryo-lamellae suitable for cryo-ET. This approach allows preparation of thin sections from virtually any part of a sample, including dense tissues and complex cellular structures.

Yet, the underlying technical constraints of cooling system stability and sample manipulation continue to limit routine adoption of these methods. Despite having successfully implemented the above workflows, we had to overcome major hurdles to apply these methods to target specific cell wall structures in plant roots.

Here, we present protocols allowing us to extend cryo-FIBSEM sessions from 24 hours to 5 days without interruptions. In addition, we describe a modified silver-plated EasyLift™ needle that eliminates the need of a copper or gold block between the original tungsten needle and the sample. Moreover, we describe a strategy that significantly reduces curtaining effects. Finally, we report a precise routine to target a lamella with a precision of approximately 1 μm in X, Y, and Z.

Together, these modifications make cryo-lift-out techniques more accessible for routine structural biology applications on any type of tissue. Our significant workflow improvements considerably reduce contamination risk and preparation time and render these powerful techniques much more accessible, significantly broadening the applicability of the techniques to solve new biological questions that were previously out of reach.

## Results

By performing cryo-ET in different developmental stages of the *Arabidopsis* root, we show that this technique allows for a much better preservation and visualization of cell wall ultrastructure, revealing different organization and orientation of the fibrils depending on the identity of the adjacent cells. We also show unprecedented details about the plasma membrane-cell wall interface, where abundant EVs appear to form, fuse with each other, and the plasma membrane during cell wall formation. We also demonstrate that various observations based on chemical fixation can be seen as valid, as they are preserved in cryo-ET material, excluding that important prior observations were due to fixation or staining artifacts.

### Meristematic cells

Tomograms from cryo-lift-out of root meristematic cells generally showed the best structural conservation with the least degree of ice crystals formation[11] (Fig. 1a–c, Supplementary Movie 1 and 2). This can be explained by the high cytoplasmic density, absence of large central

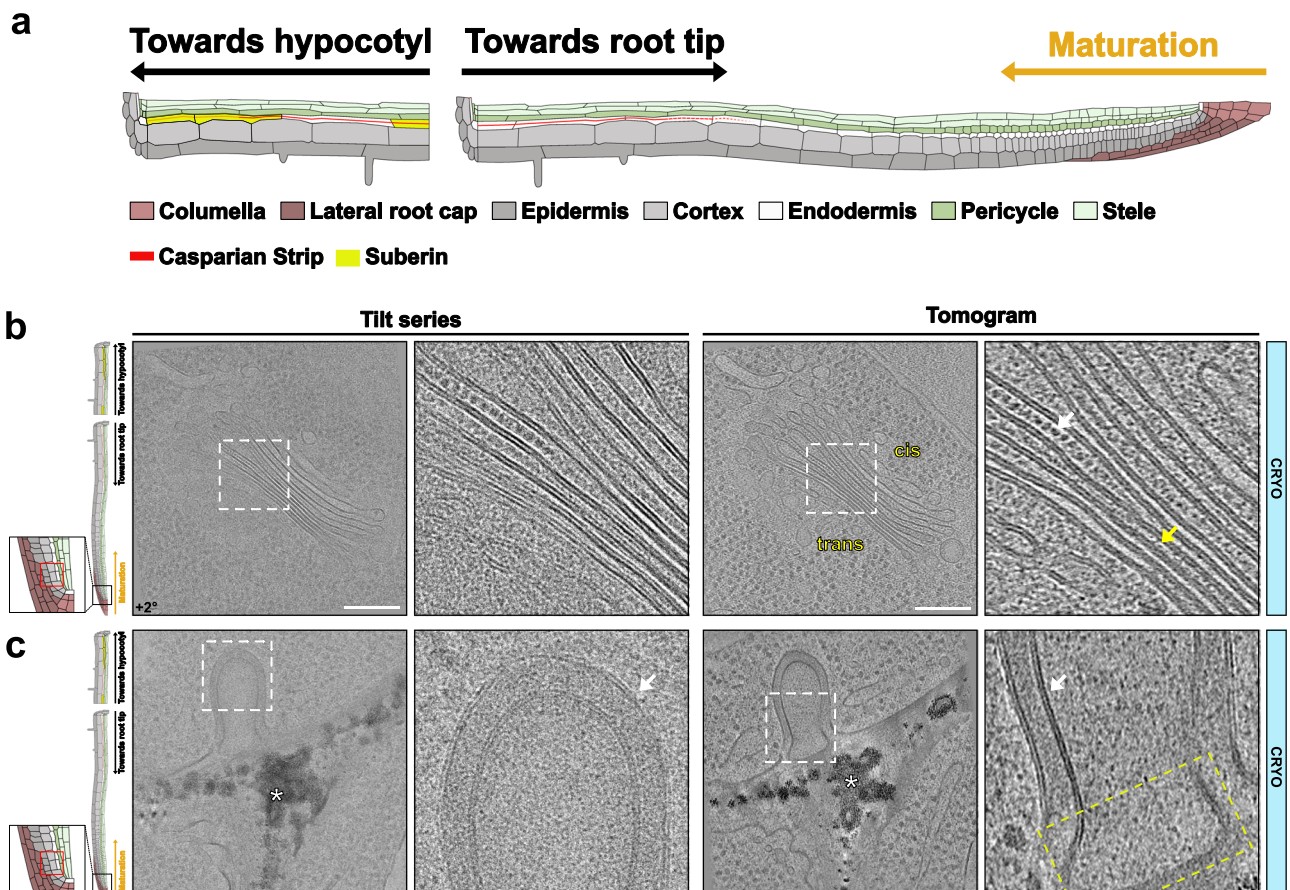

**Fig. 1 | Root tip cryopreservation and organelles identification. a** Overview of *Arabidopsis thaliana* root, indicating differentiation gradients and cell types. Localization from which images have been taken is shown based on this schematic for all figures. **b** Example of images obtained from the root tip. Golgi apparatus. White arrow – inter-cisternal elements. Yellow arrow – intra-cisternal elements (Supplementary Movie 1). **c** Example of images obtained from the root tip. Presence of a large extracellular vesicle (EV) with invagination of the plasma membrane.

White asterisk - Electron-dense precipitates (Supplementary Movie 2). Similar tomograms have been observed from 3 independent roots and sample preparations. White arrow – well-preserved plasma membrane. Yellow rectangle – highlights the stamp-like shape of the extracellular membrane. The corresponding angle from the tilt-series is shown on the images. Scale bar = 200 nm. Zoom-in field of view of 250 nm.

vacuoles, and thinner, less rigid walls of these undifferentiated actively dividing cell populations in the root tip. Figure 1b and Supplementary Movie 1 show an example of a single plant Golgi apparatus, next to a nucleus with a nuclear pore. Note the high density of ribosomes, with a clearly visible ribosomal exclusion zone and lateral differentiation of cisternae into center and periphery[11]. Moreover, a clear cis-to-trans polarity is observed with narrower cisternae towards the trans-side. More strikingly, regularly structured, inter-cisternal elements are seen, thought to be involved in the tethering of Golgi cisternae into stacks[12]. While they have been previously described in Golgi apparati of Maize, Nicotiana and Arabidopsis by HPF and cryosubstitution in root tip cells[13–15], cryo-ET allows us to observe that they are of regular size and spacing along one dimension of the cisternae and appear to be arranged helically along the z-axis, based on apparent rotation of the elements when moving along the z-axis. However, no inter-cisternal tubular connections are observed. We also compared tissue preservation in samples prepared by HPF followed by freeze-substitution (HPF/FS) as it is also a technique used to study plant ultrastructure. Despite being both cryo-fixed by HPF, the cryo-ET workflow preserves more details than the freeze-substitution approach (Supplementary Fig. 1a).

Even in meristematic cells, chemical fixation in classical transmission electron microscopy (TEM) protocols induces protoplast shrinkage, leading to plasma membrane detachment and invaginations[16]. Due to these artifacts, regularly observed extracellular membrane structures in classical TEM[16] appeared as very heterogeneous structures that were often considered artifacts. It is therefore interesting that we often observe smoothly shaped extracellular membranes in our cryopreserved samples (Fig. 1c, Supplementary Movie 2). In this example, a large extracellular vesicle, or tubule, is observed at a corner between three cells, associated with an invagination of the plasma membrane, which clearly surrounds the extracellular membrane. The origin of such structures in plants remains unclear, but they could arise from intracellular double-membrane structures, such as autophagosomes, multi-vesicular bodies, or various other less-known compartments such as EXPO[17,18]. Interestingly, the stamp-like shape of the extracellular membrane suggests that it becomes appressed against the wall, possibly due to the pressure exerted by the protoplast. Such flattened structures are also something we observe in extracellular membranes during CS formation (see below). Other features repeatedly observed in TEM images are electron-dense, precipitation-like structures within the plant cell wall. Such structures were difficult to interpret, since precipitates of this type can easily arise from heavy metals used for staining in room temperature TEM. We were therefore intrigued to repeatedly observe such precipitates in our cryo-fixed lamellae in the absence of any added heavy metals (Fig. 1c, Supplementary Movie 2). Due to their enhanced presence in cell corners and in positions near the middle lamella (see below), we speculate that these structures might arise from the presence of the high-levels of calcium in the pectin-rich regions of the cell wall.

## Unmodified cell walls in differentiated cells

For clarity, we will first describe unmodified cell walls of differentiated cells before focusing on specific cell wall modifications. Despite increased ice crystals formation in cytosol and vacuoles of differentiated cells (Figs. 2a, 3a), we could observe well-conserved rough ER cisternae, multi-vesicular bodies, and other compartments (Fig. 2a, Supplementary Movie 3). Below the plasma membrane at the cell corner, we could see well-conserved cortical arrays of microtubules (Fig. 2a, Supplementary Movie 3). The interface between the plasma membrane (PM) and cell wall showed good conservation in our tomograms. Notably, we observed aligned, parallel fibrils in the wall proximal to the plasma membrane, probably representing highly ordered arrays of cellulose microfibrils. In chemically fixed, osmium stained, and resin embedded samples imaged at room temperature TEM as well as in cryosubstituted material after HPF, we could never observe the same degree of details or ordered arrangement of microfibrils (Fig. 2b, Supplementary Fig. 1b). At best, walls in TEM pictures display a disorganized, vaguely fibrous structure, invariably appearing to have shrunk during dehydration. For adequate comparison, we have generated a tilt-series of resin-embedded sections at the

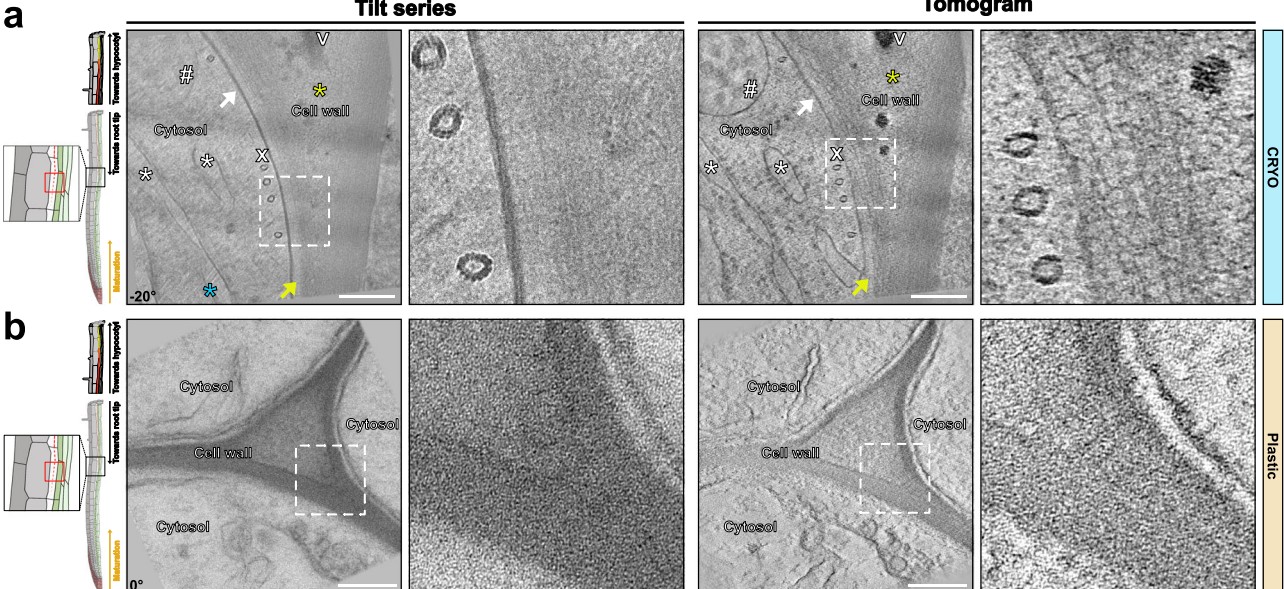

**Fig. 2 | Cell wall differentiation in the early Casparian strip (CS) formation zone at approximately 1 mm from the root tip. a** Cell wall preservation in cryo-conditions (Supplementary Movie 3). Yellow asterisk - pectin-rich region composed of disorganized fibrils of lower density. White # - multivesicular body. White V - Electron-dense precipitates. White x - arrays of microtubules. White arrows - plasma membrane. Yellow arrows - electron-lucent layer in between the fibrillar primary cell wall and the PM. Similar tomograms have been observed from 8 independent roots and 5 sample preparations. **b** Similar area that shown in (**a**) for chemically fixed and resin-embedded samples. Similar tomograms have been observed from 2 independent roots and 1 sample preparation. White asterisk - ER cisternae. Blue asterisk - visible ice crystals within the sample. Corresponding angle from the tilt-series is shown on images. Scale bar = 200 nm. Zoom-in field of view of 250 nm.

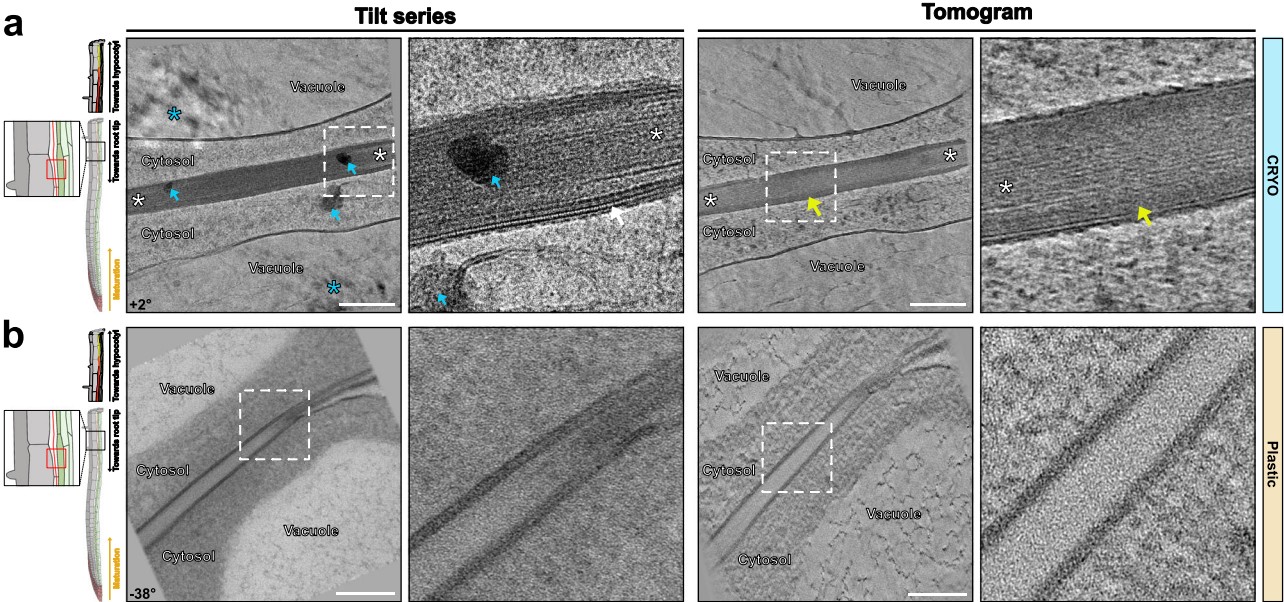

**Fig. 3 | Tomograms of the Casparian strip (CS). a** Example of cryo-fixed samples of the established CS (Supplementary Movie 4). White arrows – repetitive structural elements observed at the CS. Yellow arrows – membrane-proximal electron-dense region. White asterisk – unmodified cell wall. Blue asterisk – visible ice crystals within the sample. Blue arrow – visible ice crystal contamination present on the surface of the lamella. Similar tomograms have been observed from 8 independent roots and 5 sample preparations. **b** Similar area that shown in (**a**) for chemically fixed and resin-embedded samples. Similar tomograms have been observed from 2 independent roots and 1 sample preparation. The corresponding angle from the tilt-series is shown on the images. Scale bar = 200 nm. Zoom-in field of view of 250 nm.

same resolution as the cryo-ET (Fig. 2b). In the distal parts of the wall (towards the middle of the wall, more distant from the PM), the cell corner showed more disorganized microfibrils of lower density, consistent with the known, pectin-rich nature of this wall region. As observed in the meristematic cells, electron-dense precipitates were associated with cell corners and rarely observed in cell wall areas with dense and organized microfibril arrays. This further supports the idea that these precipitates are derived from pectin-associated calcium. Figure 2a also shows the endodermal cells having a homogenous, electron-lucent layer between the fibrillar primary cell wall and the PM. This is not observed in the cortical cells. Such differences in spacing between PM and the primary wall could not have been interpreted in room temperature TEM, as it resembles the frequently occurring artifact of PM detachment induced by fixation[16]. We now suspect that such layers are associated with cell walls that are being actively synthesized, remodeled or modified.

### Differentiated Casparian strips

To demonstrate the power of our cryo-CLEM workflow, we decided to attempt the generation of cryo-lamellae by fluorescence marker-guided cryo-lift-out to obtain tomograms of Casparian strip (CS). CS begins to appear at about 800 μm from the root tip in a 5-day-old seedling with a root of 100-150 mm in length. Moreover, within the 80–120 μm thick root cylinder, CS are exclusively formed in a single root cell layer, the endodermis, and only occupy 2–3 μm thick region along the Z axis. Reliably positioning and identifying this region is only possible due to the CASP1-GFP marker, based on a strongly endodermis-expressed protein that strictly co-localizes with CS from their moment of inception[19]. Using our cryo-CLEM workflow, we always obtained cryo-lamellae containing the CS (3 experiments). Despite ice crystal formation in the vacuole (Fig. 3a), the CS and their associated PM appeared well-preserved (Fig. 3a, Supplementary Movie 4). Again, cryo-ET delivered significantly greater details than resin-embedded and HPF/FS samples (Fig. 3b, Supplementary Fig. 1c). In TEM images of resin-embedded samples, the CS appeared as a homogenous amorphous structure. Additionally, in these conditions, the CS can be

identified as the region where plasmolysis-induced PM detachment is absent, due to the strong CASP-mediated wall adhesion of the CS membrane domain[20]. The PM at the CS, moreover, appeared to be more electron-dense than the rest of the PM, which is also observed in HPF/FS (Fig. 3b, Supplementary Fig. 1c). By contrast, cryo-lamellae revealed a PM with repetitive structural elements, possibly representing the polymeric CASP scaffold and their associated proteins[19,21,22] (Fig. 3a, Supplementary Movie 4). In addition, a membrane-proximal, electron-dense region, approximately four times the thickness of the PM itself, was observed which might be due to the high density of cell wall enzymes associated with lignification during CS formation, such as peroxidases, dirigent proteins (DIRs), and others[22–24]. Finally, we were surprised to see that, despite a clearly increased contrast of the CS, certainly due to dense lignification of their cell wall (Fig. 3a, space between asterisks), the regularly spaced fibrillar arrangement of unmodified walls was still observable. This fits with similar observations in xylem vessels (see below). Cell wall areas with CS had to be imaged at lower beam intensity, because they were more sensitive to beam damage ("burned") than other areas of the lamella. Since the same applies to xylem vessels, we assume that enhanced sensitivity to the beam is due to the presence of lignin.

### Nascent Casparian strips

Even more challenging than identifying CS for cryo-lift-outs is the generation of cryo-lamellae of the so-called string-of-pearls stage of initial Casparian strip formation[19]. The string-of-pearls stage occurs in a precise developmental window of about 1–2 cells among the more than a hundred endodermal cells present in a 5-day-old seedling and again requires precise Z-axis positioning within the root cylinder. Previous TEM analysis of this stage had revealed the presence of large numbers of extracellular vesiculo-tubular membranes[16]. The morphology of these extracellular membranes was extremely heterogeneous, probably due to the strong sensitivity of these structures to protoplast shrinkage during fixation (Fig. 4b, d). This contrasts with a more coherent shape and density of the extracellular membrane observed in cryo-conditions (Fig. 4a, c), with fields of vesiculo-tubular

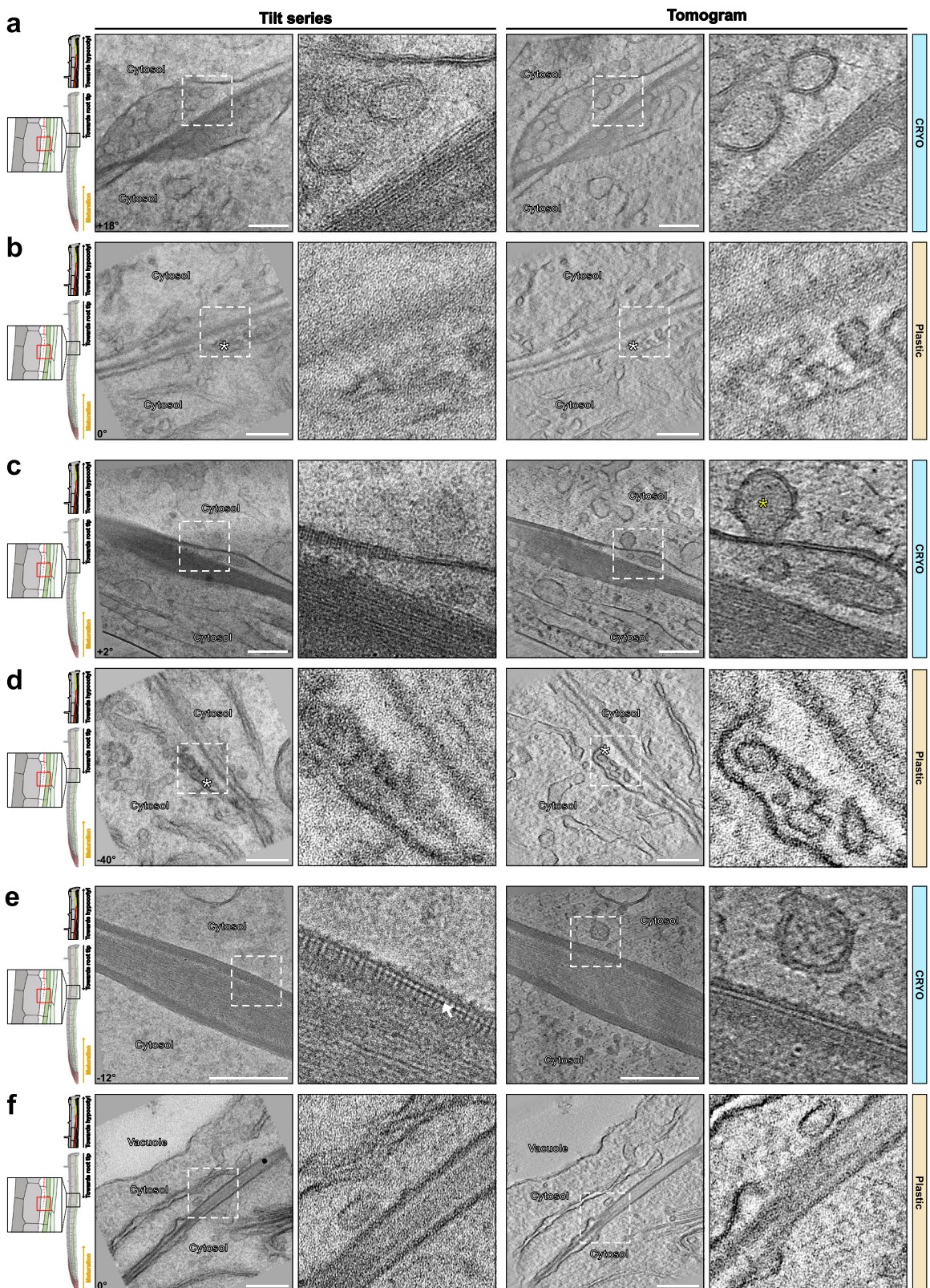

structures always residing in a lens-shaped space between the plasma membrane and the very thin, fibrous cell wall separating two endodermal cells. Tomograms of the string-of-pearls stage expectedly give divergent pictures, with either a fully formed CS with little or no extracellular vesicles (Fig. 4e, Supplementary Movie 7), an unformed CS with many vesicles (Fig. 4a, Supplementary Movie 5), or an intermediary stage with partially formed CS and fewer extracellular vesicles (Fig. 4c, Supplementary Movie 6). This is consistent with current

models, according to which the string-of-pearls stage represents partially formed CS domains, with intercalated areas of strong secretion where the CS has not yet formed[25]. The ontogenesis of the extracellular vesicles during CS formation is unclear, but one straightforward explanation could be that they are the result of an excessive membrane surface that is resolved by extrusion of PM into an extracellular space between PM and the primary cell wall. Indeed, vesiculo-tubular membrane structures tethered to the PM on its cytosolic face can easily be

**Fig. 4 | Tomograms of Nascent Casparian strip. a, c, e** Example of cryo-fixed samples of nascent CS. Similar tomograms have been observed from 3 independent roots and sample preparations. **b, d, f** Similar area that shown in (**a**), (**c**), and (**e**), respectively, for chemically fixed and resin-embedded samples. Similar tomograms have been observed from 2 independent roots and 1 sample preparation. **a** Vesiculo-tubular structures in a lens-shaped space between plasma membrane and the very thin, fibrous wall section between two endodermal cells. **b** Similar structure in resin sections showing tubular structures close to membranes, but difficult to distinguish from artefactual membrane invaginations due to plasmolysis. White arrow - flattened vesicles against the primary cell wall (Supplementary Movie 5). **c, d** Intermediary stage with partially formed CS and fewer extracellular vesicles. White asterisk - vesiculo-tubular membrane structures tethered to the PM on its cytosolic side (Supplementary Movie 6). **e, f** Fully formed CS without extracellular vesicles (Supplementary Movie 7). White arrow – nanostructures repeated within the membrane. The corresponding angle from the tilt-series is shown on the images. Scale bar = 200 nm. Zoom-in field of view of 250 nm.

observed in this region (Fig. 4b, zoom-in). An intriguing question is the eventual fate of these extracellular membranes, since they are not present anymore upon CS completion (Fig. 4e, Supplementary Movies 4 and 7). Interestingly, we could observe extracellular membranes that appeared to become appressed, flattened against the primary cell wall, suggestive of turgor pressure-driven compression and eventual resorption of these structures back into the cell (Fig. 4e, zoom-in). Very young CS regions reveal an even clearer nanostructure of the CS and its membrane domain (Fig. 4e, Supplementary Movie 7). As described above, the PM often displays very strictly ordered repeats within the membrane (Fig. 4e, zoom-in), associated with a more amorphous layer immediately outside of the PM, followed by a fibrous region of the wall, similar to unmodified primary wall, except for an increased width. Remarkably, no distinct middle lamella region can be observed within the primary cell wall. Despite resembling unmodified primary walls, the CS always appeared as slightly more electron-dense.

## Suberin lamellae formation
While also occurring exclusively in the endodermis within a primary root, suberin lamellae (SL) become deposited around the entire cell, in contrast to the very restricted formation of a CS band in the median position of endodermal cells. Suberin is chemically distinct from the lignin present in CS and occurs at a later stage of its development. It consists of hydroxylated fatty acids, which can become esterified to each other, but also to phenolics, such as ferulic acid, as well as glycerol. Despite a good knowledge of its monomeric composition, much remains unknown concerning its polymeric connections and the resulting spatial arrangements within the wall and how it is connected to unmodified or lignified primary cell wall[26]. Especially, there is still no widely accepted model that can explain the intriguing, lamellated appearance of suberin, as well as its positioning and integration between the cellulosic/lignified cell wall and the plasma membrane. We identified areas for cryo-lift-outs and lamellae preparation using a fluorescent reporter for the transcriptional activity of GLYCEROL-3-PHOSPHATE sn-2-ACYLTRANSFERASE 5 (GPAT5), a central and suberin-specific enzyme in the suberin biosynthetic pathway[27]. Suberin lamellae (SL) can be identified on tomograms of cell walls between an endodermal cell and its neighboring cortical cell, since only the cell wall on the endodermal side will display suberin formation. As described above, the unmodified cell walls showed a high degree of order with fibrous elements in parallel and equally spaced arrangements (Fig. 5a, c, e, f, Supplementary Movie 8-11). We were at first confused to repeatedly observe the previously described precipitates, nicely aligned in the cell wall space between cortex and endodermis, but always "off-center", i.e. closer to the endodermal side (Fig. 5a, c, e, f, Supplementary Movies 8–11). We now interpret this as the result of increased wall synthesis of the cortical cell as compared to the endodermal cell. This would place the middle lamella, the pectin and calcium-rich layer of attachment between plant cells, closer to the endodermal side. In the space between primary endodermal walls and the plasma membrane, we could observe a thin layer of material, not present on the cortical side. Towards the PM, the suberin layer becomes wavy, or jagged, but remains separated from the plasma membrane by an additional, membrane-proximal layer of different density. We interpret the wavy/jagged aspect of the membrane-facing side of the suberin layer as partially-formed lamellae (Fig. 5a, c, e, f, yellow arrows). Indeed, close inspection of the suberin layer allows to identify two or three lamellae constituting the suberin layer at this stage. It is important to point out that this is the first time that the lamellated structure of suberin is visualized in the absence of any heavy metals as contrasting agents, demonstrating that there are indeed inherent structural differences detectable not only by heavy metals within the suberin layer but also by phase contrast used in cryo-EM. Moreover, the absence of fixation and dehydration/embedding procedures allows more reliable estimations of the thickness of the lamellae in our samples (5–7 nm, Fig. 5c). Interestingly, when observed at the right tilt angle, suberin lamellae appeared with a contrast and thickness very close to that of the PM. This would fit with a model of a very regular polymeric arrangement in which one lamella could be defined by two units of hydroxy-fatty acids, parallelly aligned to each other, similar to fatty acids in the lipid bilayer of a membrane.

## Xylem vessel cell walls
Areas containing the thick cell walls of xylem vessels can be identified by their autofluorescence for cryo-lift-outs. Xylem vessels are the cellular units crucial for water-transporting vascular networks in plants and represent a type of heavily lignified cell that is very different from the endodermis. In contrast to the endodermis, lignification in xylem vessels is preceded by the formation of thick, secondary cell wall structures that are initially unmodified and rich in cellulose. This is then followed by strong lignification of the secondary wall thickening, in concert with the xylem cell undergoing programmed cell death, often leading to a boost in post-mortem lignification[28,29]. This is different than the endodermis, which stays alive during both lignification and suberization of its walls. Xylem vessels are devoid of cytoplasm and filled with xylem sap, containing dissolved minerals and comparatively little organic substances. We therefore expected heavy ice crystal formation but were surprised to see a very good degree of vitrification, both in the wall and in the vessel lumen. Again, highly organized fibrillar structures were observed in the xylem secondary walls, despite the presence of lignin, indicated by the higher electron-density of its cell wall compared to those of the neighboring cell (Fig. 6a, c, l indicates denser, lignified wall region). The neighboring cell, whose intact cytoplasm would indicate it to be a xylem parenchyma cell (Fig. 6a, c, Supplementary Movies 12, 13), displayed a much more disordered arrangement of microfibrils. Nothing resembling these fibrillar structures could be observed in comparable images of resin-embedded samples or HPF/FS (Fig. 6b, d, Supplementary Fig. 1d). Resin sections revealed the difference between xylem vessel and parenchyma walls through a higher degree of staining of the parenchyma cells, which consequently appeared more electron-dense than the lignified xylem vessel walls. The distinction between non-modified and lignified cell wall in HPF/FS is even less clear in our case (Supplementary Fig. 1d). Once again, cell wall structure seems to be strongly affected by the sample preparation arising from resin embedding, while in the cryo-lamellae, the differences in wall structure were already apparent from the very different orientation and degree of order of the fibrillar wall component (Fig. 6a, b, zoom-in).

## The cryo-lift-out process
The cryo-lift-out process typically involves several key steps. Here, we will focus on the main challenges and solutions we developed in our

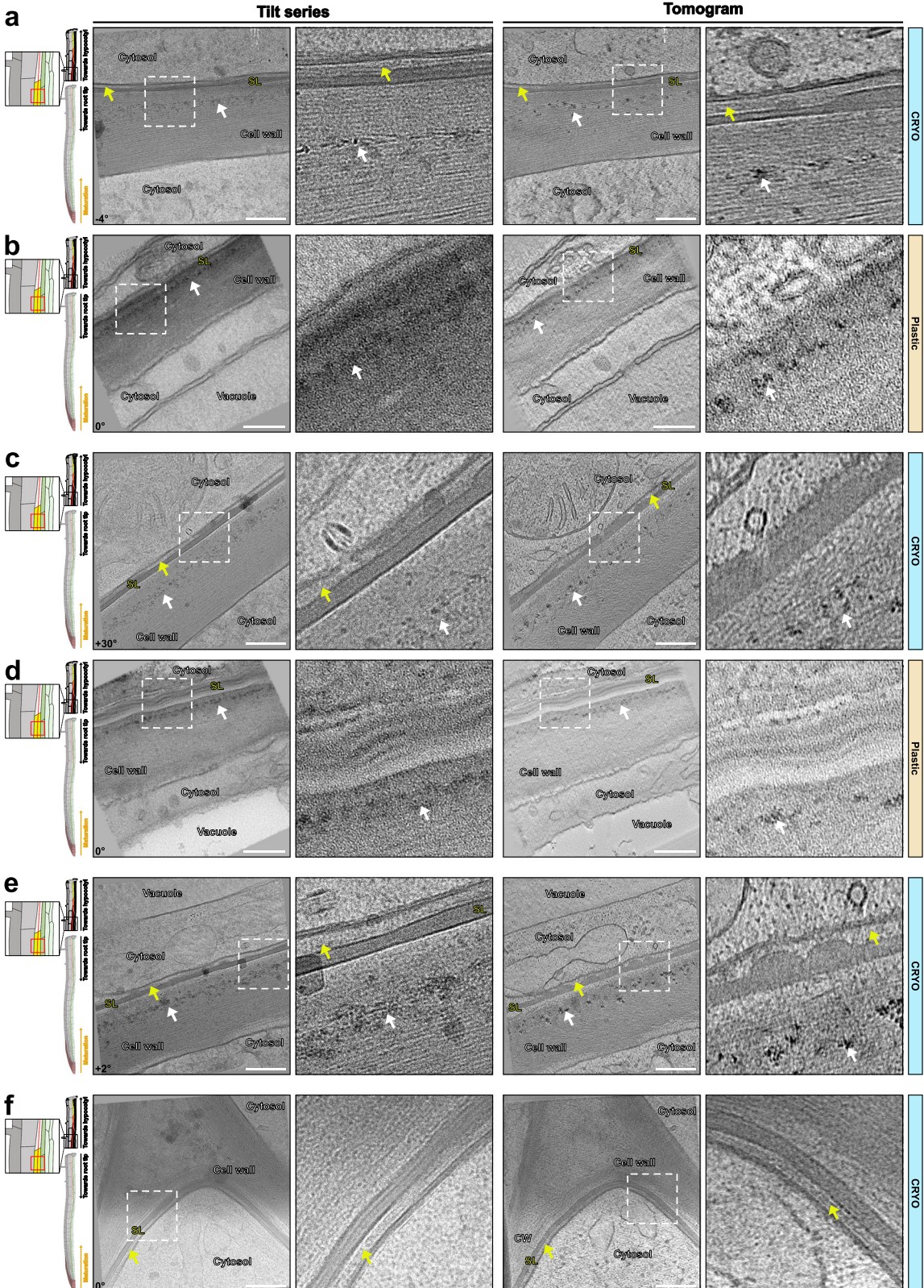

**Fig. 5 | Tomograms of mature cell wall and early suberizing cells.**
**a**, **c**, **e** Tomograms of cryo-fixed samples of early suberizing cells (Supplementary Movies 8–11). Similar tomograms have been observed from 2 independent roots and sample preparations. **b**, **d** Similar area that is shown in (**a**) and (**c**), respectively, for chemically fixed and resin-embedded samples. Similar tomograms have been observed from 2 independent roots and 1 sample preparation. **f** Tomograms of cell wall and early suberin deposition at the endodermal cell corner. SL – suberin lamella. White arrows – electron-dense precipitates. Yellow arrows – electro-lucent layer in between PM and suberin lamellae. The corresponding angle from the tilt-series is shown on images. Scale bar = 200 nm. Zoom-in field of view of 250 nm.

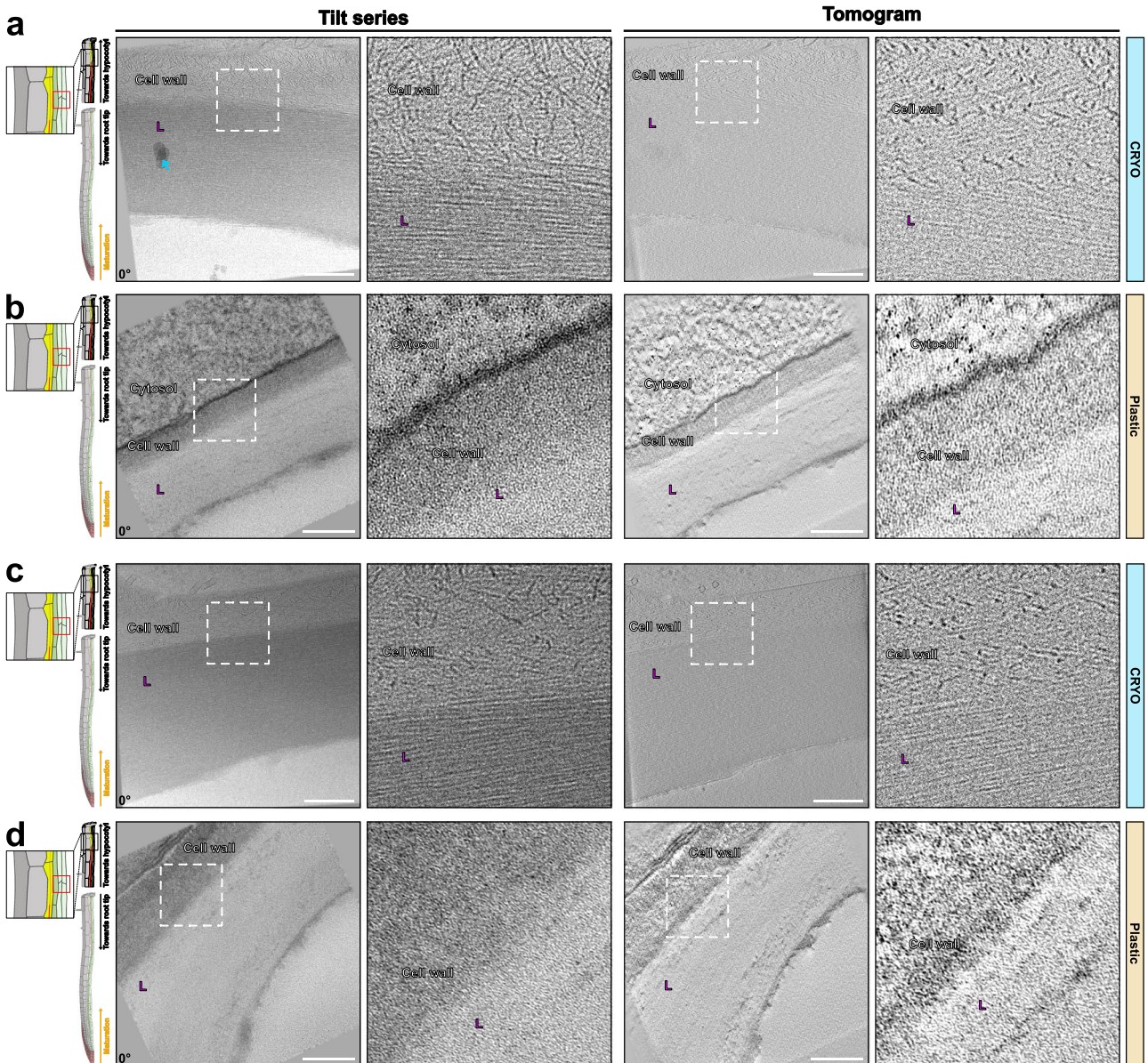

**Fig. 6 | Tomograms of lignin-impregnated cell wall of xylem vessels.**
**a**, **c** Tomograms of Xylem vessels in Cryo fixed samples (Supplementary Movies 12-13). Similar tomograms have been observed from 1 root and sample preparation. **b**, **d** Similar area that is shown in (**a**) and (**c**), respectively, for chemically fixed and resin-embedded samples. Similar tomograms have been observed from 1 root and sample preparation. Magenta L – lignin impregnated cell wall from xylem cells. Blue arrow – visible ice crystal contamination present on the surface of the lamella. The corresponding angle from the tilt-series is shown on the images. Scale bar = 200 nm. Zoom-in field of view of 250 nm.

attempts to initially target the CS. The complete workflow used to reach proper targeting will be presented. CS were visualized using the CASP1-GFP marker under its endogenous promoter[19]. Roots of the transgenic lines were rapidly frozen using HPF to preserve their native structure. After having tried different protocols, the best freezing results, minimizing ice crystal formation and cell osmotic perturbation, were obtained by cutting the tip of *A. thaliana* root, transferring it rapidly to 20% dextran inside a 100 μm deep carrier of 3 mm diameter, and immediately high-pressure freezing it (Fig. 7a).

**Targeting the Casparian strip with a precision of approximately 1 μm in XY and Z by a 2-step targeting process**
Once the root tip is vitrified in the carrier, the first step is to trim the surface by using a cryo-ultramicrotome equipped with a cryo-trim knife. This allows us to reach the appropriate distance in Z of the region of interest (ROI) as well as to smooth the vitrified sample

surface for visualization of the fluorescent signal. The carrier is then transferred to a cryo-Light Microscopy (cryo-LM) equipped with a cryo-stage. To insert our sample into the cryo-LM, the holder has been modified to accommodate the carrier (Fig. 7b). The entire surface of the carrier is imaged in the fluorescence channels, as well as reflected light, creating a map of the whole surface (Fig. 7c). Z stacks of the different ROI are also obtained at a resolution of 1 μm Z steps to determine where the cryo-lift-out block will be carved.

The vitrified sample is then transferred to the cryo-FIBSEM system. The FIB can be precisely controlled to mill a specific ROI. In the case of the root tip, images of the entire surface obtained by Cryo-LM are transferred in the software of the cryo-FIBSEM and used as a map to identify the different ROI. Since the cryo-FIBSEM used in these experiments is not equipped with an in-chamber light microscope, the targeting relies only on the perfect overlay of fluorescent maps on the cryo-FIBSEM images and precise measurements.

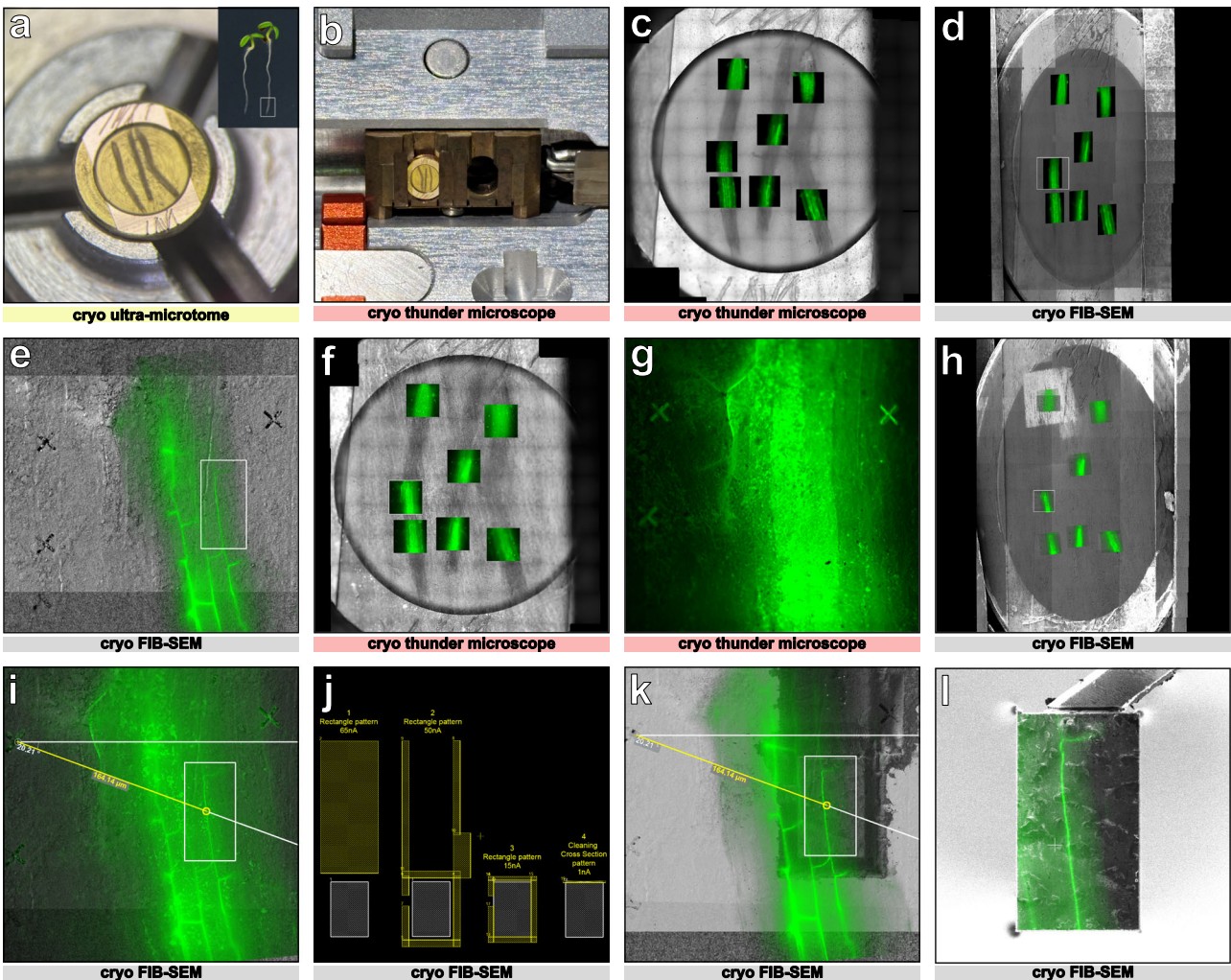

**Fig. 7 | Coarse and fine targeting using fluorescence Z-stacks and fiducial marks. a** Carrier containing three root segments in the cryo-ultramicrotome chamber. The carrier surface is polished with a diamond knife, and fiducial marks (scratches) are engraved on the carrier rim using a diamond tip. **b** Carrier mounted in a modified cryo-cassette for imaging in the cryo-Light microscope (cryo-LM). **c** Overview map of the carrier in reflective light showing the localization of the fluorescent Z-stacks of the CASP-GFP signal acquired in the cryo-LM. **d** Full map of the carrier acquired with the ion beam in the cryo-FIBSEM using MAPS software (TFS) overlaid with the fluorescent Z-stacks. **e** Enlarged view of the white square in (**d**) showing fiducial crosses milled at 100 μm of distance around the ROI. **f** Reflected light overview map of the carrier showing the fluorescent Z-stacks from the second round of cryo-LM with milled fiducial crosses. **g** Enlarged view of the white square in (**f**) with the crosses at the surface of the carrier. **h** Full map of the carrier acquired with the ion beam in the cryo-FIBSEM overlaid and accurately

aligned using the fiducial crosses of the fluorescent Z-stacks. **i** Enlarged view of the white square seen in (**h**) showing a maximum intensity Z projection containing the crosses and the target aligned with the ion beam snapshot. Polar coordinates measure the angle and distance of the target from the cross. **j** Four sequential milling patterns (yellow) are used to prepare the block (55 ×80 μm white rectangle). **k** Block prepared with the ion beam, overlaid with the fluorescent target. The block preparation includes a long opening trench at 65 nA, trenches around the block at 50 nA and then at 15 nA, undercut at 5 nA with 10° milling angle, and needle attachment face polishing at 1 nA. **l** Block attached to the silver needle through silver redeposition overlaid with the fluorescent target. The block width is reduced to 40 μm in width and the bottom is polished at 5 nA. The presented workflow has been performed 8 times with 3 roots per carrier, and at least 2 regions of interest per root were targeted.

To improve the CS targeting, we developed a routine in two steps. First, the cryo-LM map of the entire surface of the carrier is overlaid with a map of the entire surface of the carrier obtained by scanning with the ion beam. Then, the two maps are overlaid and the root tips identified (Fig. 7d). Three small crosses acting as fiducial marks are carved with the FIB at the surface of the sample surrounding ROIs (Fig. 7e). Once done, the carrier is brought back to the cryo-LM and precise Z stacks containing fiducial marks are acquired for each ROIs (Fig. 7f, g, Supplementary Movie 14). Then, the carrier is reloaded into the cryo-FIBSEM. The precise distances in X, Y and Z between the ROI and each of the fiducial marks are measured on the fluorescent images (Fig. 7 h, i) and reported on the cryo-FIBSEM images (Fig. 7k). The fiducial marks are used to determine the size and depth of the block

that will be prepared for cryo-lift-out (Fig. 7j). The typical size of a block is 50 x 80 μm and 30 μm in depth (Fig. 7l).

## Extended operation through enhanced cooling system design
To enable extended cryo-FIBSEM operations, we developed a copper spiral cold trap that integrates into the existing microscope cooling system (Fig. 8a, b). The trap consists of a manually coiled copper tube, providing a larger inner diameter (4 mm) and an additional cooling line length (70 cm) (Fig. 8a). The larger diameter of the copper tube prevents clogging while maintaining efficient cooling. The trap is connected to the standard cooling lines (Fig. 8b). The design enables compact integration within the microscope's heat exchanger housing without compromising accessibility for refilling the cooling dewar.

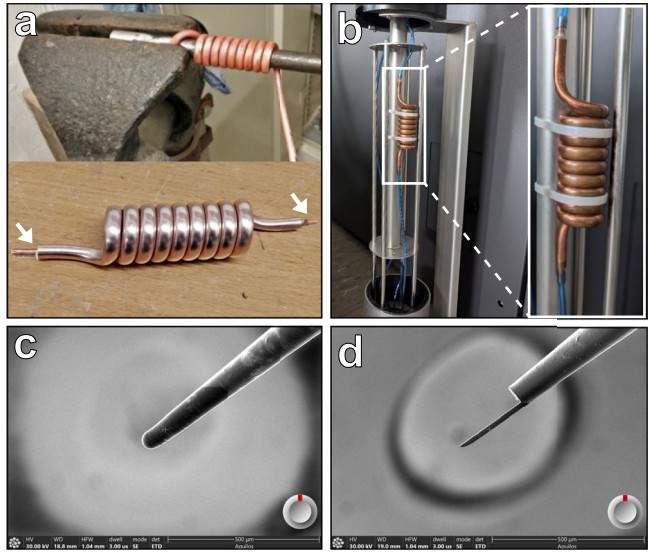

**Fig. 8 | Implementation of copper spiral cold trap and silver coated EasyLift™ needle. a** Fabrication of the cold trap showing manual coiling of a 70 cm long copper tube (6 mm outer diameter, 4 mm inner diameter) around a 16 mm steel bar to create the spiral geometry. Two tubes (arrow), 1 cm long and 4 mm outer diameter, are soldered at both ends for connection into the cooling system (arrow). **b** Installation of the cold trap in the microscope's heat exchanger, showing connection with the existing blue cooling lines. **c, d** Views of the silver-plated needle from the ion beam (IB) before and after milling, respectively. The bottom right circle with the red mark indicates the EasyLift™ rotation.

Performance tests demonstrated that the spiral cold trap extends continuous operation from the previous 12–24 hours limit to over 5 days without system clogging. This enhancement enables the completion of entire sessions without interruption. The system maintains stable cryo-conditions, eliminating the risk of ice contamination of the sample during transfer and thermal cycling of the microscope during multi-day sessions. The cooling dewar is refilled twice per day.

### Silver-plated needles for direct sample manipulation

To address the time-consuming nature of adaptor chunk preparation[8,9], we developed a silver-plated needle system enabling direct sample attachment (Fig. 8c, d). The needles are prepared through electroplating of standard tungsten EasyLift™ needles in a silver cyanide bath (Supplementary Fig. 2). Two fine-silver anodes and precise control of voltage and current ensure homogeneous plating. The plated needles undergo a systematic preparation protocol utilizing the EasyLift™ rod's rotational capability (Supplementary Fig. 3). We verified that 90° clockwise and counterclockwise rotations could be performed under both ambient and cryo-conditions without compromising vacuum and needle temperature integrity. This rotation enables precise shaping of the needle through a series of milling steps (Supplementary Fig. 4). The final needle measures 250 μm in length and is thinned to a final thickness of 17 μm in both dimensions, giving a 25 μm horizontal attachment surface. While initial preparation requires approximately 4 hours, the resulting needle can be used for months of operation.

### Enhanced sample attachment performance

The silver needle enables direct attachment to the sample blocks without intermediate adaptor chunks (Fig. 9a, Supplementary Fig. 5). Silver's higher sputter rate compared to copper facilitates robust attachment through single-pass regular cross-section milling. We demonstrated successful cryo-lift-out and attachment of sample lamellae between grid bars (Serial Lift-Out, Supplementary Fig. 5a–d) or onto silver film supports (SOLIST, Fig. 9b, Supplementary Fig. 5e–h).

This approach eliminates the 23 minutes chunk preparation previously required. The simplified workflow involves a single welding step rather than the traditional dual welding process, decreasing the risk of sample loss during cryo-lift-out and serial sectioning (Supplementary Movies 15, 16).

### Enhancement of lamella stability through attachment to silver-coated support films

As a receiver grid for SOLIST lamellae we used standard 200 mesh copper R2/2 Quantifoil grid sputter-coated with silver to obtain a film thickness of 500 nm. The thick metal layer enabled the attachment of the lamellae using single pass regular cross-section pattern by redeposition of silver on the two sides of the lamella (Fig. 9c–e). The attachment strength was tested by pushing the lamella using the silver needle with 10 steps of 200 nm (Supplementary Movie 17). This strong attachment and the film thickness make a stable system, which enables very stable polishing steps. We can typically polish 20 μm wide lamellae without lamellae bending, up to a thickness of 150-200 nm.

### Lamella preparation for curtaining reduction

The curtaining propensity during focused ion beam milling is predominantly influenced by the surface topography of the milling front. While chemical vapor deposition of organometallic platinum via gas injection system (GIS) can help planarize the surface prior to milling, its effectiveness is contingent upon the initial smoothness of the substrate. Organometallic platinum deposition on a roughly milled surface results in inhomogeneous protective layers with localized discontinuities that are readily apparent in cryo-TEM imaging (Supplementary Fig. 6a). These heterogeneities in the protective layer contribute significantly to differential milling and subsequent curtaining artifacts in the final lamella. Although the conventional GIS angle of incidence (30°) may contribute to incomplete filling of surface features, even perpendicular GIS deposition does not fully eliminate these discontinuities (Supplementary Fig. 6b). Through systematic optimization, we established a protocol that substantially reduces curtaining. Notably, low-current polishing of the milling front without initial thin organometallic platinum deposition on the top of the lamella proves insufficient for curtaining reduction. The sequential application of a thin organometallic platinum layer on the top of the lamella, followed by a front surface polishing, and a second organometallic platinum deposition on the front of the polished lamella minimizes curtaining artifacts (Fig. 9e–h). The enhanced control over lamella geometry afforded by the serial lift-out method enables precise manipulation of the milling front topography, thereby optimizing protective layer uniformity and ultimately reducing curtaining in the final lamella (Fig. 9h). This approach represents a significant improvement, providing a reproducible method for high-quality lamella preparation with minimal curtaining artifacts (Supplementary Fig. 7).

## Discussion

Unlike animals, plants have evolved complex multi-cellularity based on walled, pressurized cells, allowing plants to rapidly grow and maintain cellular structure by generating large internal vacuoles and high osmotic pressure. This internal pressure generates turgidity because it is resisted by intricately structured cell walls. Cell wall nanostructure is dynamically modified by plant cells during growth, development, and adaptation to environmental challenges. A plethora of cell-type-specific cell wall modifications are crucial for allowing many plant cells to carry out their specific function: Lignin impregnations lend rigidity to secondary xylem cell walls and render primary endodermal cell walls impermeable to solutes, suberin lamellae are added below primary cell walls as a protective, hydrophobic coating of the protoplast, whereas cutin is deposited outside of epidermal cell wall for the

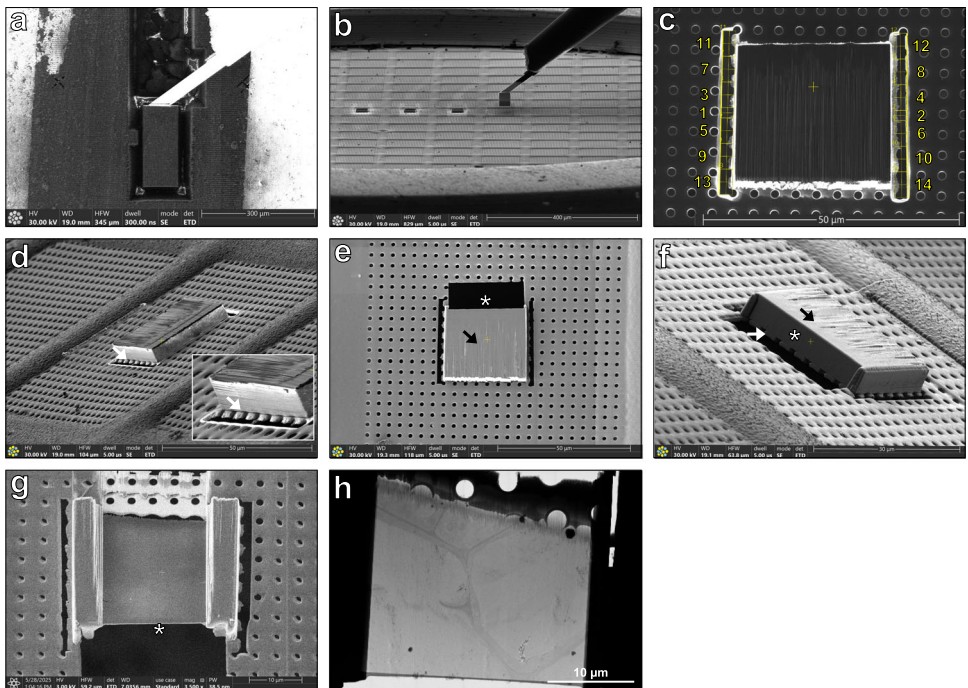

**Fig. 9 | Lamella preparation using the silver needle and the SOLIST method with reduction of curtaining. a** Direct attachment of silver needle to the sample block from a HPF carrier using single-pass regular cross-section milling. **b** Sample block lift-out after successful lamella deposition on silver support film. **c** Lamella (width 30 μm, height 27 μm, thickness 5 μm) perpendicular to the ion beam showing single-pass regular cross-section milling patterns (yellow rectangles) for attachment on both sides of the lamella. The numbers from 1 to 14 indicate the milling sequence. **d** Lamella at 10 degrees to the ion beam and rotated (R -110°) showing the silver redeposition on the side of the lamella (white arrow) making a strong attachment to the silver film. **e** Same lamella then seen in (**c**) and (**d**), perpendicular to the ion beam. Front surface polishing (asterisk) at high current (1 nA) after initial organometallic platinum deposition (20 seconds) by gas injection system (GIS) perpendicular to the top surface of the lamella (black arrow). **f** Lamella at 10 degrees to the ion beam and rotated (R −30°) showing the lamella front surface (asterisk) after polishing at 1 nA and additional organometallic platinum deposition on the polished front surface (asterisk). The black arrow indicates the top surface of the lamella with the organometallic platinum layer, and the white arrow indicates the thick silver film. **g** Optimized prepared lamella exhibiting a continuous, void-free organometallic platinum protective layer and exceptionally smooth lamella surface without curtaining artifacts after polishing steps. Imaged with the electron beam at 3 kV showing very low charging effect on the lamella. A very thin organometallic platinum layer remains at the front of the lamella (asterisk). **h** Example of Cryo-lamella in Cryo-TEM with curtaining reduction.

same purpose. Thick cellulosic walls contain the high osmotic pressure of phloem sieve elements, while localized thinning of walls occurs during elongation growth in root hair, for example. Cell walls provide structure and coherence to plant tissues and have rendered them a favorite object of investigation since the beginning of microscopy[30]. However, these same features have also rendered plant tissues a challenge for cryo-ET, since large vacuoles, pressurized cells, and rigid cell walls, impede efficient vitrification. Vacuoles increase the likelihood of ice crystal formation, which can severely compromise sample integrity and resolution. Furthermore, the high internal pressure and associated rigidity of the plant cell also limit the tolerance of the tissue to mechanical constraints, especially osmotic stress, which can rapidly induce severe artifacts, especially at the crucial plasma membrane-cell wall interface. Therefore, while it is commonly accepted to add cryoprotectant to improve vitrification of animal samples, we use cryoprotectant only to fill the carrier some seconds before freezing to avoid osmotic artifacts. Impermeability of cell walls can also reduce the even penetration of cryoprotectants, complicating efforts to achieve complete vitrification. While complete vitrification remains a technical hurdle that needs to be solved, our protocol demonstrates significant improvements in sample handling and structural preservation, making previously inaccessible plant tissues amenable to cryo-ET analysis. Despite these challenges, we successfully imaged Casparian strips, as well as suberized cell walls in cryo-lamellae, demonstrating the power of our enhanced cryo-CLEM workflow. Casparian strips are highly localized, ring-like impregnations of primary cell walls with hydrophobic lignin, generating a critical, extracellular diffusion barrier across the endodermis, allowing the root to regulate solute and water transport. Later, endodermal cells further suberize, forming so-called suberin lamellae between the plasma membrane and primary cell wall on all endodermal surfaces, fully sealing off the endodermal cell from its environment with this hydrophobic, cork-like secondary cell wall. It was indeed the observation of a field of highly suberized, dead cells that Robert Hooke observed in sections of cork tissue, and which led to his coinage of the term "cells"[30]. Our work now visualizes endodermal suberin and lignin structures in near-native conditions and allows an unprecedented examination of their architecture and formation. Our data reveal an unexpectedly high order and spatial regulation of fibrils in the cell wall, with surprising degrees of parallel arrangements distal to the plasma membrane and more disorganized arrays closer to the PM. While the CS PM domain has previously been reported to be more electron dense and to display tight attachment to the plasma membrane, cryo-ET now reveals the CS-associated plasma membrane to form a much thicker, electron-dense layer than previously observed, possibly representing the dense accumulation of extracellular and transmembrane proteins known to localize to this domain. The unmodified primary cell wall of endodermal cells, consisting of cellulose microfibrils, hemicellulose, and pectins, also displays a degree of parallel order that we do not observe in conventional TEM. In the cell corners, a much more disorganized arrangement of fibrils is observed. Furthermore, cryo-ET of suberin in situ provides the first visualization of lamellated suberin in the absence of any EM staining and embedding that could have altered its native structure. This demonstrates that suberin lamellae are indeed

made of polymers of different electron densities and do not simply have differing affinities for EM stainings.

Our improvements in cryo-lift-out techniques—including the use of silver-plated EasyLift™ needles and an extended cryo-FIBSEM operational workflow—have improved targeting, while ensuring the generation of many lamellae of high quality from a region of interest. These advancements significantly enhance the reproducibility and feasibility of cryo-ET for many previously challenging or inaccessible tissues or organs.

While our method relies on automation software such as AutoTEM™ Cryo for lamella preparation and polishing, a significant portion of the workflow remains manual. Critical steps, including block preparation for lift-out, lamella attachment to grids, front-face polishing, and organometallic platinum deposition are currently not supported by automation software and therefore done manually. Development of software solutions allowing automation of such tasks is of great importance to improve and facilitate cryo-FIBSEM lamellae generation.

In summary, while vitrification remains a persistent challenge, the findings presented here represent a significant advancement in cryo-electron microscopy. Further refinements of vitrification protocols and cryo-ET methodologies will continue to push the boundaries of structural plant biology, offering deeper insights into the fundamental processes governing root function and environmental adaptation.

To conclude, our study demonstrates that recent advancements in cryo-ET significantly enhance the structural analysis of plant root tissues while maintaining consistency with traditional chemical fixation methods. The ability to preserve native cellular structures with minimal artifacts offers insights into plant cell architecture and molecular organization. Importantly, our findings confirm that cryo-ET does not invalidate prior models but rather refines them, allowing for a more detailed examination of cellular processes such as suberin deposition and Casparian strip formation.

By comparing full cryo-workflows with resin embedding protocols, we highlight the importance of selecting the most suitable technique based on specific research questions. Cryo-ET excels in preserving osmotic balance and ultrastructural integrity, enabling high-resolution investigations of plant cell wall dynamics. However, traditional fixation remains valuable for large-scale and quantitative analyses.

As cryo-ET methodologies continue to evolve, they will provide unprecedented opportunities to explore plant cell biology at the molecular level. Future studies integrating correlative light and electron microscopy approaches will further bridge the gap between structural and functional analyses, advancing our understanding of plant development and adaptation.

## Methods

### Room temperature TEM analysis
Samples were prepared as described previously[16]. Plants were fixed in glutaraldehyde solution (EMS, Hatfield, PA) 2.5% in phosphate buffer (PB 0.1 M [pH 7.4]) for 1 h at RT and postfixed in a fresh mixture of osmium tetroxide 1% (EMS, Hatfield, PA) with 1.5% of potassium ferrocyanide (Sigma, St. Louis, MO) in PB buffer for 1 h at RT. The samples were then washed twice in distilled water and dehydrated in ethanol solution (Sigma, St Louis, MO, US) at increased concentrations (30–40 min; 50–40 min; 70–40 min; 100%–2x 1 h). This was followed by infiltration in Spurr resin (EMS, Hatfield, PA, US) at graded concentrations (Spurr 33% in ethanol - 4 h; Spurr 66% in ethanol - 4 h; Spurr 100%–2x 8 h) and finally polymerized for 48 h at 60 °C in an oven. For electron tomography, semi-thin sections of 250 nm thickness were cut. The area of interest was taken with a transmission electron microscope JEOL JEM-2100Plus (JEOL Ltd., Akishima, Tokyo, Japan) at an acceleration voltage of 200 kV with a TVIPS TemCamXF416 digital camera (TVIPS GmbH, Germany). Micrographs were taken as single tilt series

over a range of −60° to +60° using SerialEM[31] at tilt angle increments of 1°. Tomogram reconstruction was done with IMOD software[32].

### High pressure freezing and freeze substitution
Root segments (5 mm) from Arabidopsis thaliana were placed in 100 μm deep cavities of 6 mm aluminum carrier (Art.610, Wohlwend GmbH, Switzerland) filled with 15% dextran and 1% trehalose dissolved 2-morpholioethanesulfonic acid buffer (MES 50 mM, pH5.7). The carrier was covered with a tap carrier (Art.611, Wohlwend GmbH, Switzerland) and vitrified using a high-pressure freezing machine HPF Compact 02 (Wohlwend GmbH, Switzerland). The roots were then freeze-substituted using the Leica AFS2 freeze-substitution machine (Leica Mikrosysteme GmbH, Vienna, Austria). Dehydration and fixation was done in a solution containing a mixture of osmium tetroxide 0.5% (EMS, Hatfield, PA) with glutaraldehyde 0.5% (EMS, Hatfield, PA) with uranyl acetate 0.1% (Sigma, St. Louis, MO) in acetone (Sigma, St. Louis, MO, US) at graded temperature (−140 °C to −90 °C for 1 h; −90 °C for 30 h; from −90 °C to −60 °C in 6 h; −60 °C for 10 h; from −60 °C to −30 °C in 6 h; −30 °C for 10 h; from −30 °C to 0° in 6 h) This was followed by washing in acetone and then infiltration in Spurr resin (EMS, Hatfield, PA, US) at graded concentration and temperature (30% for 10 h from 0 °C to 20 °C; 66% for 10 h at 20 °C; 100% twice for 10 h at 20 °C) and finally polymerized for 48 h at 60 °C in an oven.

### High-pressure freezing
Root segments (2 mm) from Arabidopsis thaliana were placed in 100 μm deep cavities of 3 mm gold-coated copper carriers (Type A, Wohlwend GmbH, Switzerland) filled with 20% (w/v) dextran (Sigma, St. Louis, MO) dissolved in MOPS buffer. The carrier was covered with a Type B carrier coated with hexadecene (Sigma, St. Louis, MO) and vitrified using a Leica EM ICE high-pressure freezer (Leica Microsystem, Vienna, Austria). After freezing, carriers were stored in cryobox in liquid nitrogen until further processing.

### Cryo-ultramicrotomy
The carrier was transferred to a Leica UC6 ultramicrotome (Leica Microsystem, Vienna, Austria) cooled to −170 °C, and the surface was trimmed using a Trim20 diamond knife (Diatome, Biel, Switzerland). After complete polishing of the surface, fiducial marks (scratches) were engraved on the carrier rim using a diamond tip (EMS, Hatfield, PA, US) to facilitate later alignment and correlation (Fig. 7a). The carrier was then stored in a cryobox in liquid nitrogen.

### Cryo-correlative light and electron microscopy workflow
For the first acquisition in the cryo-LM, the carrier was mounted in a modified cryo-cassette (Fig. 7b) for the Leica Thunder cryo-LM (Leica Microsystem, Vienna, Austria) and transferred to the cryostage for imaging. A complete overview map (11 x 12 tiles) of the carrier was acquired using reflected light with a 50x objective (NA 0.75, pixel size 260 nm), using four focus points set to ensure sharp focus across the entire surface. Fluorescence z-stacks of the CASP1-GFP signal were collected using a GFP filter set (EX: 450-490, DC 495, EM: 500-550). The Z-stacks were initiated at the carrier surface and extended below the cells of interest with a step size of 1 μm (Fig. 7c). The carrier attached to the cryo-cassette was mounted on a 45° pretilt shuttle, gently cleaned with a fine brush if ice contamination was present, and transferred to the Aquilos 2 cryo-FIBSEM (Thermo Fisher Scientific Inc., US). Remnant of ice contamination was removed quickly using a 5 nA ion beam at fast scan (50 ns, 768 x 512) at milling position. The carrier was then sputter coated with platinum at 1 kV, 30 mA, 0.15 mbar for 15 seconds. The center of the carrier was set to eucentric height, rotated to 110° and tilted to 7°, perpendicular to the ion beam. A full carrier montage was acquired using the ion beam in Maps™ software (Thermo Fisher Scientific Inc., US), with 6 × 11 tiles, overlap of 10% in X and 50% in Y, tile HFW of 950 μm, 1536 x 1024, dwell of 1μs, 4 frames, and three focus

points ensuring sharp focus of the carrier surface. The resulting ion beam map exhibited after stitching, an elongation along the vertical axis, a known limitation of MAPS software when using ion beam imaging. Fluorescence data was imported into Maps™ and aligned to the ion beam image using three reference points (Fig. 7d). Despite the image distortion along the vertical axis, the overlay enabled direct stage navigation to regions of interest (ROIs). Because each montage has small local misalignments due to the stitching process, aligning the two montages together can result in a mislocalisation of the target up to 15-20 μm. To overcome this problem, three cross-shaped fiducial markers were milled at 100 μm of distance from each ROI using a 1 nA ion beam current (crosses: 2 x rectangular pattern, X 20 μm, Y 2 μm, Z 0.2 μm) (Fig. 7e). This distance should be sufficient to avoid damage to the sample while keeping the fiducials within the field of view of the cryo-LM. Once all the targets were surrounded by fiducial crosses, the carrier was then transferred back to the Leica Thunder cryo-LM. The carrier was imaged again as described above, but the acquired Z-stacks now included the milled fiducial crosses (Fig. 7f, g). These images would serve as the foundation for precise three-dimensional targeting of the regions of interest. Following imaging, the carrier was removed from the modified cassette and stored in a cryobox in liquid nitrogen.

## Silver support film for SOLIST lamella deposition

For SOLIST implementation, we used standard R2/2 Quantifoil carbon film on 200 mesh copper grids (Quantifoil Micro Tools GmbH, Germany) sputter-coated with 500 nm of silver using a Safematic CCU-010 sputter coater (Safematic GmbH, Switzerland) at an argon pressure of $5 \times 10^{-2}$ mbar and a process current of 30 mA. Grids are placed at 5 cm from the silver target, giving a coating rate of 0.4 nm/seconds, requiring 20 minutes to reach the desired thickness of 500 nm.

The grid was then clipped in a standard Autogrid with a C-clip (Thermo Fisher Scientific Inc., US), and three reference marks were made on the Autogrid rim with a permanent pen: two dots aligned with the grid bars defining the future tilt axis and a third dot, on the side of the Autogrid, perpendicular to the tilt axis.

## Fine targeting in cryo-FIBSEM

The receiver silver grid was loaded at room temperature into the 45° pre-tilt shuttle under a binocular microscope, with the grid bars horizontally aligned such that the two reference dots were positioned horizontally (at 3 and 9 o'clock) and the third dot at the bottom (6 o'clock). This precise alignment facilitated subsequent serial sectioning of the lifted material.

The shuttle was then cooled down in the cryostation, where the carrier was loaded and aligned using the engraved diamond marks (scratches) and gently cleaned with a fine brush and liquid nitrogen if ice contamination was present. After loading into the Aquilos™ 2 FIBSEM, an ion map was acquired following the procedure described in the Cryo-CLEM microscopy workflow section, and the fluorescent data were imported and aligned (Fig. 7h).

To avoid distortions from the elongated Z stacks and achieve precise targeting, the fluorescence Z stacks were reimported a second time, and the stage was navigated to each ROI. Ion beam snapshots were taken at each position, clearly visualizing the fiducial crosses, and the corresponding Z stacks were aligned using two alignment points, avoiding distortion. Polar measurements (angle and distance) between fiducial crosses and features of interest were drawn in Maps™, and these precise X-Y targeting coordinates are then drawn in xT software to accurately place the milling pattern to create the block (Fig. 7i, k).

## Axial-scaling factor to accurately target in Z

For accurate Z targeting, we accounted for the refractive index mismatch between the sample medium (water, $n = 1.33$) and the immersion medium (N₂ gas, $n = 1$). Using established correction factors for the Leica Thunder microscope (NA = 0.75, GFP emission $\lambda = 509$ nm), we determined that the actual depth of the target (e.g., 14 μm) was 1.46 times greater than its apparent depth. An additional safety margin of 10 μm below the calculated target depth was implemented to protect the sample during undercutting.

## Block preparation

These preparation steps assume that the silver needle is already prepared following the Silver-Plated Needle Preparation, the EasyLift™ rod rotation check, and the silver needle milling steps. The ROI was centered on screen using Maps™ software, the crosses are localized, and the precise coordinates of the block were defined using polar measurement from one cross. The target depth (eg. 14 μm) was corrected using the 1.46 axial factor (Corrected target depth = TD = 14 x 1.46 = 20.44 μm) plus the 10 μm safety margin. An opening trench was milled above the ROI, with the opening length (OL) calculated as: OL = (TD + 10)/tan (MA). The milling angle (MA), defined as the angle at which the undercut is performed beneath the block to release it from the carrier. For a corrected target depth of 20.44 μm, this resulted in an opening length of approximately 170 μm. The trench was milled using a 65 nA beam current. A secondary milling pattern was then applied at 50 nA to cut around the block, leaving a connecting bridge to the bulk material. The current was reduced to 15 nA when approaching the final block dimensions (50 × 80 μm) (Fig. 7j). The stage was then rotated to −70° and tilted to 17° (giving a 10° milling angle) to do the undercut of the block and the connecting bridge using a 5 nA rectangular pattern. The stage was then returned to the original position (110° rotation, 7° tilt), and the surface facing the opening trench was polished with a 1 nA cleaning cross-section pattern to make a smooth surface for needle attachment.

## Silver needle sample attachment

The silver-plated needle tip was flattened using a 5nA rectangular pattern prior to sample block attachment. The sample block surface was set at the coincident point, and the EasyLift™ needle was inserted and brought into contact with the block surface. Attachment was achieved through silver redeposition by placing a horizontal array of regular cross-sections (single pass, width 0.5 μm, lateral spacing 0.25 μm, height 2 μm, z-depth 4 μm) at 30 kV and 1nA with the milling direction toward the silver needle. The block was released from the carrier by milling through the remaining bridge with a rectangular pattern (z-depth 10 μm, 30 kV, 5 nA) and lifted out of the carrier. The needle was retracted, the stage was lowered to remove the carrier from the field of view, the needle was reinserted, and the block width was reduced to 40 μm, and the bottom was polished with a 5 nA current (Fig. 7l). Depending on the methods used (Serial Lif-Out, SOLIST) the sample is then respectively attached between grid bar (Supplementary Fig. 5d) or deposited on the silver film of the receiving grid (Supplementary Fig. 5h).

## SOLIST procedure

The SOLIST procedure was performed as described previously[9]. The first grid square for lamella deposition was positioned 6 squares to the left and 2–3 lines above the grid center. After setting the first grid square at coincidence point and 10° milling angle, the EasyLift™ needle was inserted, the block was lowered to slightly touch the silver film at the center of the grid square and a 42 μm wide, 600 nm high, and 3 μm deep rectangular milling pattern was used at 1 nA to section a 5 μm thick lamella from the block. After each cut, the block was raised, and the grid was moved by a relative displacement of −0.125 mm to the next grid square, repeating the process until the entire block was sectioned. This procedure allowed an 80 μm long block to yield approximately 13–14 lamellae.

## Lamella attachment on silver film

The grid was rotated 180° and tilted 7° perpendicular to the ion beam for attachment of all lamellae. Two arrays of regular cross-section (single pass, width 2 μm, height 5 μm, z-depth 0.7 μm, 30 kV, 1 nA) were placed on the silver film on both sides of the lamella with the milling direction toward the silver film. Depending on the size of the lamella, one array can contain 4 to 8 milling patterns. The milling patterns were numbered following a milling sequence where the 1st pattern is milled at the left of the lamella, the 2nd at the right, the 3rd at the left etc… to create a uniform, balanced, and strong attachment of the lamella to the silver film.

## Lamella preparation to reduce curtaining

Once all the lamellae were attached, the gas injection system (GIS) was inserted perpendicular to the grid, and the flow was opened for 20 seconds to cover the top surface of each lamella. The front surface of the lamellae was then polished using cleaning cross-section patterns at 30 kV, 1 nA (z-depth 4 μm). The grid was then returned to the 10° milling position, and the GIS was reinserted for a 60 seconds deposition to cover the polished front surface of each lamella.

## Lamella thinning

Final thinning was performed using AutoTEM™ Cryo software (Thermo Fisher Scientific Inc., US) starting at 1 nA to achieve an initial lamella thickness of 2 μm. Progressive thinning continued in steps: 1.4 μm thickness at 0.5 nA, 800 nm at 0.3 nA, 500 nm at 0.1 nA. Final polishing was performed manually at 50 pA using parallel rectangular milling patterns with an overtilt of 0.4–0.8° to thin down the back side of the lamella and achieve uniform thickness.

## Cryo-Electron Tomography

Data acquisition was performed on a Titan Krios G4 instrument operated at 300 kV equipped with a Selectris™ X energy filter and a Falcon 4i camera (Thermo Fisher Scientific Inc., US). Tomographic data were collected using the Tomo5 software package (Tomo5 v5.2, Thermo Fisher Scientific Inc., US). Lamella-overview montages were acquired at a magnification of ×11,500 (pixel size, 2.236 nm). Tilt series were recorded at a magnification of ×53,000, with a pixel size of 2.42 Å and stored in EER file format. Data collection followed a dose-symmetric tilt scheme with 2° angular increments, with 3 e−/Å² per tilt and a target defocus of −4.5 μm. The angular range spanned from −60° to 40°, resulting in a total accumulated dose of 150 e−/Å².

## Tomogram reconstruction

Frames were summed to get at least 0.8e-/px and motion corrected with the alignframes command from IMOD package version 5.0.1[32]. Tomogram reconstruction was done with Etomo from IMOD package version 5.0.1.

## Cold Trap Design and Implementation

A 1 m long copper tube, 6 mm outer diameter, 4 mm inner diameter, was annealed with a torch and quenched in tap water to soften it. It is then attached strongly in a vice with a 16 mm cylindrical steel bar and manually rolled up around it to create a spiral (Fig. 8a). Two smaller brass tubes of 4 mm outer diameter were brazed at the two ends of the copper spiral with silver solder. It is then cleaned with 10% sulfuric acid in water for 10 min, then rinse several times with tap water and brushed with a brass brush to remove the oxidation layer. It is then cleaned with deionized water and dried with acetone. The cold trap is then mounted in the heat exchanger of the cooling dewar by cutting and removing a small portion of the existing blue tube (blue tube length to remove = cold trap length – 2 small brass tube) and attach firmly to the main tube of the heat exchanger with two nylon cable ties as seen in Fig. 8b. During operation, the cold trap functions passively and doesn't require any maintenance. The procedure below should be performed at the conclusion of extended microscope sessions, typically after 5 days of continuous operation to prevent trapped water from migrating into the microscope stage during system warm-up:

- Discontinue nitrogen gas flow.
- Extract the heat exchanger from the cooling dewar.
- Apply localized heating with a heat gun to the bottom tip of the cold trap to release the blue tubing connection.
- Restore nitrogen gas flow at 200 mg/s and heat the entire cold trap until complete moisture removal.
- Reestablish the blue tubing connection to the cold trap.
- Reduce nitrogen gas flow to 10 mg/s for stage warming.

This maintenance protocol ensures safe system warm-up while preserving the microscope stage performance for subsequent sessions.

## Silver-plated needle preparation

EasyLift™ tungsten needles were silver plated using an RNG-1502 power supply (Bijoutil, Geneva, Switzerland) (Supplementary Fig. 2), which maintains a constant voltage/current and allows a continuous movement of the needle, maintaining optimal plating conditions. The needle is first attached to an electric conductive holder and plunged in a degreasing bath containing Electrolytic degreasing type "A" (Bijoutil, Geneva, Switzerland) at a concentration of 70 g/L in H2O, using two stainless steel anodes under a constant voltage of 2.5 V and a current of 0.2 A for 1 min at RT. It is then rinsed several times in tap water, followed by rinsing in deionized water. The needle is then plunged in the silver cyanide solution "Argent W brillant" (Finishing, La-Chaux-de-Fonds, Switzerland) at a silver concentration of 25–40 g/L using two fine silver anodes (Bijoutil, Geneva, Switzerland) under a constant voltage of 0.5 V and a current of 0.1 A for 10 min at RT. The needle is then rinsed several times in tap water, followed by rinsing in deionized water and finally dried at room temperature. A thickness measurement check was done under a binocular using a 300 square mesh copper grid (space between the grid bars is around 50-60 μm) attached with a transparent tape on a glass slide. The needle tip was aligned onto the grid to estimate the thickness by comparing it with the width of a grid square.

## EasyLift™ rod rotation check

The 90° rotation check of the EasyLift™ rod was done at room temperature with the microscope chamber open. In the starting position, the red mark of the EasyLift™ connector should be upward (Supplementary Fig. 4), and the cooling braid clamp should be on the right. Then, the EasyLift™ rod was slowly rotated clockwise (CW) and continuously checked to see if the copper cooling braid had enough freedom to reach 90° of rotation (red mark of the EasyLift™ connector to the right, Supplementary Fig. 3) without tension on the cooling braid and its clamp (Supplementary Fig. 3). Then the rod was rotated back counterclockwise (CCW) slowly to 0° (red mark upward Supplementary Fig. 3) while checking the movement of the cooling braid. This rotation should not put too much tension on the cooling braid, and the clamp should stay in place while rotating.

Once this rotation check was done, the cooling braid clamp was opened, the EasyLift™ rod was removed from the microscope to install the silver needle, and finally reinstalled in the microscope. The rotation of 90° CW/CCW was then done under high vacuum and cryo-condition for leak checking (the vacuum value of the chamber can be checked when rotating 90° CW/CCW).

## Silver needle milling steps

The preparation can be performed at room temperature or under cryogenic conditions. The needle is inserted and moved to the coincidence point between the electron beam (EB) and the ion beam (IB) (Supplementary Fig. 4a, b). A polygonal pattern is drawn at the needle

tip to remove the top half of the needle material over a length of 250 μm (Z-depth 40 μm, 30 kV, 65 nA) (Supplementary Fig. 4 c–e). Using the EB view the silver layer thickness around the tungsten core was measured and should be around 25-30 μm (Supplementary Fig. 4e). A rectangular tilted pattern is drawn below the needle to creates a flat bottom face over a length of 250 μm (height 1 μm, Z-depth 40 μm, 30 kV, 15 nA) (Supplementary Fig. 4f). While scanning with the IB (1536 x 1024, 200 ns, 10 pA) the EasyLift™ rod is rotated 90° CW until the red mark of the EasyLift™ connector is positioned to the right, exposing the bottom face of the needle in the IB view. Two polygonal patterns are drawn to thin down the needle to 25 μm (Z-depth 40 μm, 30 kV, 65 nA) (Supplementary Fig. 4f, g). The EasyLift™ rod was then rotated 90° CCW to the original position by placing the red mark of the EasyLift™ connector to the top, and a polygonal pattern was drawn to remove the remaining tungsten and achieve 17 μm thickness (Z-depth 40 μm, 30 kV, 15 nA) (Supplementary Fig. 4h). The EasyLift™ rod was rotated 90° C,W and two rectangular tilted patterns were drawn to thin down the needle to 17 μm in the second dimension (Z-depth 40 μm, 30 kV, 15 nA) (Supplementary Fig. 4i, j). The EasyLift™ rod was finally rotated 90° CCW to its original position. The final needle had a length of 250 μm and 17 μm thickness in both dimensions, given a horizontal width of 25 μm for attachment to the sample (Supplementary Fig. 4k, l).

## Data availability
All Source data that support the findings of this study are available from the corresponding authors upon reasonable request.

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

## Acknowledgements

This work is based on the publications of two Nature Methods papers that have inspired and guided us during this work. We thank the authors of these key papers for the detailed protocols that allow us to implement the technique on the EM facility and motivate us to develop them. We thank Alexander Myasnikov and Bertrand Beckert from the Dubochet Center of Imaging, UNIL and EPFL, Lausanne, for their training and

supervision for cryo-ET acquisitions. We thank Christian Zimmerli for fruitful discussions during the development of the method. We thank Christopher Thompson (Thermo Fisher Scientific) for the discussion about metal sputter rates that led us to the idea of making the silver needle. We thank the company Delmic, which has loaned the Ceres clean station that was instrumental in reducing the ice contamination during our transfers. This work was supported by SNSF grants (10002702 and 197739) to N.G.

## Author contributions

J.D., E.B., N.G. and C.G. designed the study. J.D. designed and implemented the technical improvements. E.B. coordinated the experiments involving the plants. J.D., E.B. and D.D.B. performed the experimental studies. J.D., E.B., D.D.B., N.G. and C.G. analyzed the data and wrote the manuscript.

## Competing interests

The authors declare no competing interests.
