## [Transparent Peer Review file · Nature Communications]

Targeted imaging of specialized plant cell walls by improved cryo-CLEM and cryo-electron tomography

Corresponding Author: Professor Niko Geldner

Version 0:

Reviewer comments:

Reviewer #1

(Remarks to the Author)

This manuscript describes a pipeline to obtain cryo-lamellas from high-pressure frozen Arabidopsis roots inside freezing planchettes using a lift-out approach. The pipeline includes fluorescence detection of markers at cryogenic temperatures to identify regions of interest, extended operation of the cryo-FIB SEM through a copper spiral cold trap, and silver-plated needles for cryo-lamella lift-out. As a proof of concept, the authors present cryo-electron tomographic data of Arabidopsis roots, with a special focus on walls developing Casparian strips. The authors highlight that Arabidopsis roots are relatively large and difficult to handle for cryo-ET studies. This technique could facilitate the cryo-milling process of large samples, not only plant roots.

The tomographic data presented, though, provides very limited new insights into Casparian strip deposition or plant cell biology in general. For the most part, the cryo-ET data is contrasted with images of chemically fixed roots visualized by conventional TEM. This is hardly useful, considering that electron tomography of high-pressure frozen roots processed by cryo-substitution and resin embedded has been widely used to analyze plant cells. This technique has shown Golgi stacks and tubular structure resembling EXPO organelles as described in Figure 1. I am not implying that cryo-ET has no value here but comparing cryo-ET data to chemically fixed tissues does not provide a fair assessment of the power of cryo-ET in this case. The data about Casparian strip structure could be interesting but again, no meaningful comparisons between available approaches is made. In addition, the cryo-ET data is presented in descriptive and vague manner, without quantifications.

In conclusion, the pipeline designed by the authors to be able to collect cryo-lamellas from high-pressure frozen samples is promising and timely, but the resulting data on plant cell architecture and cell wall formation in roots is underdeveloped.

Reviewer #2

(Remarks to the Author)

Having read this paper, it is clear that there has been a huge effort undertaken to refine and improve the tricky handling and sample preparation steps to deliver site-specific lamellae for electron tomography. The authors have chosen to focus on plant material, and by their own admission, some of the cryo sample preparation is trickier on plant materials than mammalian tissue or cells. However, they have delivered scientific insight into the workings of the Arabidopsis thaliana roots using the specialised correlative light and electron microscopy approaches that build on the work of others. This is open an admission, and what marks this paper out is how the authors go on to demonstrate some of the biggest issues that are holding this area back, which build on those works. The authors have at every step compared their sample preparation to equivalent resin-prepared samples and show that the cryo-ET data is superior, underlining the need for the sample preparation to enable the science and understanding to be unlocked.

I have nothing bad to say about this paper, save a few missing scale bars in figure 11.

The clear result of their development work undertaken to solve many issues, such as reliable sample attachment, reduction of curtaining and long run times in the electron microscope. These improvements will I have no doubt, be of use to others in the community looking to emulate this work with their own samples.

I have no hesitation in recommending it for publication.

Reviewer #3

(Remarks to the Author)

General Comments

This is a technically impressive and methodologically advanced study that introduces substantial improvements to cryo-FIBSEM workflows, particularly the Serial Lift-Out and SOLIST approaches. The authors demonstrate successful application to plant root tissues, targeting structures such as the Casparian strip, suberin lamellae, and secondary xylem walls, which are highly relevant for plant function but previously inaccessible to cryo-ET.

Noteworthy results include:

- Extension of cryo-FIBSEM sessions from 24 hours to 5 days, significantly increasing efficiency.
- A modified silver-plated EasyLift™ needle that simplifies the setup and reduces contamination risks.
- A strategy that mitigates curtaining effects, improving lamella quality.
- A precise targeting routine (~1 μm accuracy), enabling reliable access to specific ultrastructural features.
- Preservation of hydrated plant cell wall nano-structure with minimal artifacts, opening new avenues for studying wall architecture under near-native conditions.

The work is significant to plant biology, structural biology, and microscopy methodology, as it refines existing workflows and enables new biological insights. Compared with the established literature, this work represents a clear methodological advance over traditional resin-embedding and chemical fixation approaches, while complementing recent developments in cryo-ET applied to multicellular samples. The originality lies in adapting cryo-lift-out strategies specifically to the unique challenges of plant tissues.

Overall, the conclusions are well supported by the presented data, but some clarifications and additional evidence would further strengthen the manuscript (see below). The methodology is sound and meets high standards in the field, though certain technical details should be clarified to ensure reproducibility.

Specific Comments (Point-by-Point)

Originality and Significance

1. The title could be improved by specifying the type of plant cells or tissues studied, to clearly signal the biological focus to readers.

Methodology and Technical Details

2. In the abstract, the phrase “these modifications considerably reduce contamination risk and preparation time” is too general. Please specify which modifications are meant, and clarify how strategies for reducing curtaining and achieving precise lamella targeting contribute to lowering ice contamination risk.

3. High-pressure freezing: Please explain why the root segments were placed in 20% (w/v) dextran dissolved in MOPS buffer. Could this treatment influence root morphology (e.g., via osmotic stress from dextran)?

4. Cryo-CLEM workflow: Please provide details of the platinum sputter coating (voltage and thickness).

5. Materials and Methods: Were any post-processing steps applied (e.g., software-based destripping of curtaining artifacts)? If so, a brief explanation of the principle would help reproducibility.

Data and Evidence

6. Please state how many samples, grids, or cells were analyzed for the results shown in Figure 1, to ensure the robustness of the conclusions.

7. Figures 2, 3, and 6: Consider marking areas with ice contamination in vacuoles or cytosol to show the morphology of ice particles. This would better support the claim that the target cells are minimally affected by ice.

8. Cryo-Lift-Out Process: Including a comparison of different modes of ice crystal formation and osmotic perturbations would give a more direct impression of how methods differ in efficiency and preservation quality.

9. Figure 13: Please clarify what each image represents (time-based, progress-based, or a set of examples). If they are only examples, labeling them with “A” may not be necessary. Also, black dots appear in nearly all images—please explain what these are.

Interpretation and Conclusions

10. Discussion: The statement “type-specific cell wall modifications are crucial for allowing many plant cells to carry out their specific function” would be stronger if specific modifications were listed (e.g., thickening, thinning, lignification, suberization) and whether they are developmentally regulated, stress-induced, or both.

Version 1:

Reviewer comments:

Reviewer #1

(Remarks to the Author)

In their reply, the authors have answered my concerns about the comparison between chemically fixed-resin embedded samples to high-pressure frozen-ice embedded samples. My concerns were based on the fact that high-pressure freezing has been recognized (for several decades now) as the best way to preserve biological materials for TEM; much better and reliable than chemical fixation. Because of this, it is very infrequent to find electron tomography studies using chemical fixation. Thus, I appreciate that in their reply the authors included images of high-pressure frozen/resin embedded roots to

establish a more useful comparison to appreciate the many advantages of cryo-electron tomography. This comparison allows us to appreciate the value of imaging cells under cryogenic conditions, instead of remarking all the very well known flaws of chemical fixation. It is disappointing, though, that none of the new data has been incorporated into the manuscript for the readers to see.

Reviewer #3

(Remarks to the Author)

The concerns from my side have been mostly solved. One point left is the thickness of sputter coating layer in specimen, although authors have provided the voltage and pressure parameters. The reason is that coating thickness is sensitively influenced by topography and the coating layer thickness may mask fine structure if it is too thick.

Point-to-point answers to reviewers

Reviewer #1 (Remarks to the Author):

This manuscript describes a pipeline to obtain cryo-lamellas from high-pressure frozen Arabidopsis roots inside freezing planchettes using a lift-out approach. The pipeline includes fluorescence detection of markers at cryogenic temperatures to identify regions of interest, extended operation of the cryo-FIB SEM through a copper spiral cold trap, and silver-plated needles for cryo-lamella lift-out.

As a proof of concept, the authors present cryo-electron tomographic data of Arabidopsis roots, with a special focus on walls developing Casparian strips. The authors highlight that Arabidopsis roots are relatively large and difficult to handle for cryo-ET studies. This technique could facilitate the cryo-milling process of large samples, not only plant roots.

The tomographic data presented, though, provides very limited new insights into Casparian strip deposition or plant cell biology in general.

REPLY: In the results section, we added a first paragraph summarizing the new insights provided by the cryo-ET workflow we present. We consider the cryo-ET results to be in continuity with the results previously gathered with chemical fixation. This is an important finding for the community that can thus build on results observed by chemical fixation to address new questions.

“By performing cryo-ET in different developmental stages of the Casparian strip, we show that this technique allow a better preservation and visualization of the structure of the cell wall, for exemple highlighting the different organization and orientation of the fibrils depending on the identity of the adjacent cells. We also show unprecedented details about the interface where EV are fusing with the plasma membrane to form the Cell wall. We also prove that the previous observations based on chemical fixation are valid and preserved in cryo-ET material excluding that major observations were due to artefacts”

For the most part, the cryo-ET data is contrasted with images of chemically fixed roots visualized by conventional TEM. This is hardly useful, considering that electron tomography of high-pressure frozen roots processed by cryo-substitution and resin embedded has been widely used to analyze plant cells.

REPLY: We disagree about the conventional TEM to cryo-ET comparison not being useful, as it is a widely used EM technique for analysis of plants. To address the Reviewer’s comment, we are providing a number of precise comparisons of High-Pressure Frozen (HPF) / freeze-substituted (FS) samples (HPF/FS) that match specific

samples in our manuscript (**Reply Figure 1**). We hope the Reviewer will appreciate the strong differences compared to cryo-ET, notably a complete loss in the HPF/FS technique of the intricate structures of cell wall polysaccharides that we can observe in cryo-ET (**Reply Fig. 1a**), rendering this technique unsuited for the study of cell walls. **Reply Fig. 1b** shows that the overall structure of a Golgi apparatus is nicely conserved with HPF/FS, but that there is a severe loss of detail with respect to membrane structure (no bilayer structure can be observed), as well as a loss of details in the inter-cisternal space. The same applies for extracellular membrane structure (**Reply Fig. 1c**). Even more pronounced, all the intricate structures of the cell wall and difference between non- and lignified cell wall of the xylem vessels is not observable (**Reply Fig. 1d**). Similar conclusions can be made for all the different structures presented in the study by cryo-ET.

Reply Figure1. Tomograms of plant structures using HPF/FS. **a** established CS. **b** Golgi apparatus. **c** Extracellular membrane, **d** Xylem vessels. High-Pressure Freezing (HPF) followed by freeze-substitution (FS)

This technique has shown Golgi stacks and tubular structure resembling EXPO organelles as described in Figure 1.

REPLY: We are not sure about the study or studies the reviewer is referring to and would have appreciated references in this context. In the result section, we added a sentence as well as 3 references about the Golgi apparatus in plants and first descriptions of the inter-cisternae structures, mentioning cryo-substitution. When comparing our data with these publications, the cryo-ET clearly allows us to describe

the structure with more details and we preserve the intra-cisternal space as well as the inter-cisternal space (see Reply Figure 1b). These observations will be part of another study as it will require significant additional work that is out of the scope of the present study.

I am not implying that cryo-ET has no value here but comparing cryo-ET data to chemically fixed tissues does not provide a fair assessment of the power of cryo-ET in this case. The data about Casparian strip structure could be interesting but again, no meaningful comparisons between available approaches is made. In addition, the cryo-ET data is presented in descriptive and vague manner, without quantifications.

REPLY: See our comment above, we do think that comparisons with chemically-fixed samples are useful as this technique is widely used and actually can often provide much better membrane details than HPF/FS samples, despite the many fixation artifacts. We agree with the Reviewer that this study is mainly descriptive and not quantitative, something that would require many more samples for each subcellular structure described and require whole new, stand-alone studies for each of those structures. Our aim with this broad overview that we provide is to (a) demonstrate the potential of cryo-ET in revealing new details for many previously studied subcellular structures, particularly cell walls, (b) to provide highly relevant confirmation that many structures observed in chemically-fixed samples, such as extracellular membrane, electron-dense wall precipitates and others, are not due to fixation or staining artifacts, as they can also be observed in cryo-tomography. This study therefore provides a confirmatory basis on which many groups will be able to build on.

In conclusion, the pipeline designed by the authors to be able to collect cryo-lamellas from high-pressure frozen samples is promising and timely, but the resulting data on plant cell architecture and cell wall formation in roots is underdeveloped.

REPLY: We agree that our paper is a particular hybrid of a methods paper that also provides biological novelty in the form of first descriptions of important plant structures by cryo-ET and their comparison to older techniques. We maintain that the biological data provided is a powerful illustration of the significance of our technical improvements, since nobody – to our knowledge – has provided targeted generation of cryo-lamellae from Arabidopsis roots to date and since we provide an order of magnitude more cryo-lamellae than the recent publication from the Frommer and Baumeister groups (Pöge et al., BioRxiv, 2025)

Reviewer #2 (Remarks to the Author):

Having read this paper, it is clear that there has been a huge effort undertaken to refine and improve the tricky handling and sample preparation steps to deliver site-specific lamellae for electron tomography. The authors have chosen to focus on plant material, and by their own admission, some of the cryo sample preparation is trickier on plant materials than mammalian tissue or cells. However, they have delivered scientific insight into the workings of the *Arabidopsis thaliana* roots using the specialised correlative light and electron microscopy approaches that build on the work of others. This is open an admission, and what marks this paper out is how the authors go on to demonstrate some of the biggest issues that are holding this area back, which build on those works. The authors have at every step compared their sample preparation to equivalent resin-prepared samples and show that the cryo-ET data is superior, underlining the need for the sample preparation to enable the science and understanding to be unlocked.

I have nothing bad to say about this paper, save a few missing scale bars in figure 11. The clear result of their development work undertaken to solve many issues, such as reliable sample attachment, reduction of curtaining and long run times in the electron microscope. These improvements will I have no doubt, be of use to others in the community looking to emulate this work with their own samples.

I have no hesitation in recommending it for publication.

REPLY: We thank reviewer 2 for the support. We deeply appreciate their understanding of the aim of this work and its relevance. Sharing these improvements to help the entire community to use cryo-ET with similarly complicated samples is our key take home message.

Reviewer #3 (Remarks to the Author):

General Comments

This is a technically impressive and methodologically advanced study that introduces substantial improvements to cryo-FIBSEM workflows, particularly the Serial Lift-Out and SOLIST approaches. The authors demonstrate successful application to plant root tissues, targeting structures such as the Casparian strip, suberin lamellae, and secondary xylem walls, which are highly relevant for plant function but previously inaccessible to cryo-ET.

Noteworthy results include:

- Extension of cryo-FIBSEM sessions from 24 hours to 5 days, significantly increasing

efficiency.

- A modified silver-plated EasyLift™ needle that simplifies the setup and reduces contamination risks.
- A strategy that mitigates curtaining effects, improving lamella quality.
- A precise targeting routine (~1 μm accuracy), enabling reliable access to specific ultrastructural features.
- Preservation of hydrated plant cell wall nano-structure with minimal artifacts, opening new avenues for studying wall architecture under near-native conditions.

The work is significant to plant biology, structural biology, and microscopy methodology, as it refines existing workflows and enables new biological insights. Compared with the established literature, this work represents a clear methodological advance over traditional resin-embedding and chemical fixation approaches, while complementing recent developments in cryo-ET applied to multicellular samples. The originality lies in adapting cryo-lift-out strategies specifically to the unique challenges of plant tissues. Overall, the conclusions are well supported by the presented data, but some clarifications and additional evidence would further strengthen the manuscript (see below). The methodology is sound and meets high standards in the field, though certain technical details should be clarified to ensure reproducibility.

Specific Comments (Point-by-Point)

Originality and Significance

1. The title could be improved by specifying the type of plant cells or tissues studied, to clearly signal the biological focus to readers.

REPLY: We agree with Reviewer 3, but could not find a title that was much more specific without being too long. However, we took care to mention the use of Arabidopsis, as well as the different cell walls targeted in the shortened, fully re-written abstract.

Methodology and Technical Details

2. In the abstract, the phrase “these modifications considerably reduce contamination risk and preparation time” is too general. Please specify which modifications are meant, and clarify how strategies for reducing curtaining and achieving precise lamella targeting contribute to lowering ice contamination risk.

REPLY: We modified the abstract to be more specific and reduced it to meet Nature Communications format.

3. High-pressure freezing: Please explain why the root segments were placed in 20%

(w/v) dextran dissolved in MOPS buffer. Could this treatment influence root morphology (e.g., via osmotic stress from dextran)?

REPLY: We have extensively explored the conditions of HPF in the past. Here, we provide the results obtained with a figure (Reply Figure 2) showing the effects of cryoprotectants at short (5min) and long term (1h) on the root morphology by monitoring plasmolysis using confocal microscopy and a plasma membrane marker. We found that 20% dextran in MES or MOPS was the combination providing less disruption at short and long term while some cryoprotectant had dramatic effects. We do not plan to add this in the current study. We could offer to write a Nature Protocols including such aspects.

Osmotic stress - Plasmolysis assay with plasma membrane marker

Reply Figure 2. Testing of osmotic stress induced by cryoprotectants.

4. Cryo-CLEM workflow: Please provide details of the platinum sputter coating (voltage and thickness).

REPLY: Sputtering was done at 1 kV, 30mA, 0.1mbar for 15 seconds (Added in “Cryo-correlative light and electron microscopy workflow” section.)

5. Materials and Methods: Were any post-processing steps applied (e.g., software-based destriping of curtaining artifacts)? If so, a brief explanation of the principle would help reproducibility.

REPLY: All the post-processing performed is described in the paper. No post-processing has been applied to remove curtaining artefacts and no denoising has been applied.

Data and Evidence

6. Please state how many samples, grids, or cells were analyzed for the results shown in Figure 1, to ensure the robustness of the conclusions.

REPLY: Images from 3 root tips prepared independently have been acquired. From these roots, similar observation has been obtained and this on different lamellae, which means different regions/cells from these roots.

7. Figures 2, 3, and 6: Consider marking areas with ice contamination in vacuoles or cytosol to show the morphology of ice particles. This would better support the claim that the target cells are minimally affected by ice.

REPLY: We added blue asterisk in the figure showing ice crystals presence within the sample (formed during the HPF step), corresponding legends have been adapted. We also explicitly labeled the different compartments (Cell wall, Vacuole and Cytosol) on each figure for clarity. In addition, we added blue arrows, showing the presence of ice contamination, which are formed on top of the lamella while transporting and imaging it.

8. Cryo-Lift-Out Process: Including a comparison of different modes of ice crystal formation and osmotic perturbations would give a more direct impression of how methods differ in efficiency and preservation quality

REPLY: The critical step during which ice crystal formation can occur within the sample is the high pressure freezing (HPF), which depends on sample preparation/manipulation and the cryoprotectant used. Concerning the formation of ice contamination on the top of the carrier and/or the lamellae, it mainly depends on the time spent to prepare and produce the lamellae in the cryo-FIBSEM as well as when transferring the grid from the cryo-FIBSEM into the Titan Krios. Indeed, a controlled atmosphere with low humidity is critical.

9. Figure 13: Please clarify what each image represents (time-based, progress-based, or a set of examples). If they are only examples, labeling them with “A” may not be necessary. Also, black dots appear in nearly all images—please explain what these are.

REPLY: Figure 13 is now a supplementary figure and the “A” has been removed.

Interpretation and Conclusions

10. Discussion: The statement “type-specific cell wall modifications are crucial for allowing many plant cells to carry out their specific function” would be stronger if specific modifications were listed (e.g., thickening, thinning, lignification, suberization) and whether they are developmentally regulated, stress-induced, or both.

REPLY: We now added the following list of examples after this statement: *“Lignin impregnations lend rigidity to secondary xylem cell walls and render primary endodermal cell walls impermeable to solutes, suberin lamellae are added below primary cell walls as a protective, hydrophobic coating of the protoplast, whereas cutin is deposited outside of epidermal cell wall for the same purpose. Thick cellulosic walls contain the high osmotic pressure of phloem sieve elements, while localized thinning of walls occurs during elongation growth in root hair, for example.”*

Point-to-point answers to reviewers

Reviewer #1 (Remarks to the Author):

In their reply, the authors have answered my concerns about the comparison between chemically fixed-resin embedded samples to high-pressure frozen-ice embedded samples. My concerns were based on the fact that high-pressure freezing has been recognized (for several decades now) as the best way to preserve biological materials for TEM; much better and reliable than chemical fixation. Because of this, it is very infrequent to find electron tomography studies using chemical fixation. Thus, I appreciate that in their reply the authors included images of high-pressure frozen/resin embedded roots to establish a more useful comparison to appreciate the many advantages of cryo-electron tomography. This comparison allows us to appreciate the value of imaging cells under cryogenic conditions, instead of remarking all the very well known flaws of chemical fixation. It is disappointing, though, that none of the new data has been incorporated into the manuscript for the readers to see.

REPLY: We are grateful that Reviewer 1 now considers that we have answered his concerns. As suggested, we have added the reply figure 1 in the manuscript as Supplementary Figure 1 and we have modified the text accordingly to enrich the comparison. We now refer to the comparison with chemical-fixed and with HPF/FS samples.

Reviewer #3 (Remarks to the Author):

The concerns from my side have been mostly solved. One point left is the thickness of sputter coating layer in specimen, although authors have provided the voltage and pressure parameters. The reason is that coating thickness is sensitively influenced by topography and the coating layer thickness may mask fine structure if it is too thick.

REPLY: We thank Reviewer 3 to have accepted our answers. Concerning the sputter coating of the carrier, we unfortunately have no precise way to measure the thickness of the platinum. To clarify, the aim of this platinum layer is simply to create a conductive layer to avoid charges, which will improve the quality of all the manipulations done on the carrier, including generation of the fiducials for the fine targeting as well as the generation of the further lift-out block, which will contain the ROI. Moreover, the region initially sputter coated (top of the carrier) corresponds to the front of the final lamella, which is removed during lamella preparation to reduce the curtaining effect (see corresponding section) or

will serve as attachment for the silver needle, depending on the orientation of the ROI desired within the final lamella. Therefore, this first layer of platinum will never influence the structure obtained by Cryo-ET since removed during the lamellae preparation. We hope the provided explanation clarify the final concerns of reviewer 3.